

# Physical Processes Leading to Extreme day-to-day Temperatures Changes, Part I: Present-day Climate

Kalpana Hamal and Stephan Pfahl

Institut für Meteorologie, Freie Universität Berlin, 12165 Berlin., Germany

Correspondence to: Kalpana Hamal (*k.hamal@fu-berlin.de*)

**Abstract.** Extreme temperature changes from one day to another, either associated with warming or cooling, can have a significant impact on health, environment, and society. Previous studies have quantified that such day-to-day temperature (DTDT) variations are typically more pronounced in the extratropics compared to tropical zones. However, the underlying

physical processes and the relationship between extreme events and the large-scale atmospheric circulation remain poorly understood. Here, these processes are investigated for different locations around the globe based on observation, ERA5 reanalysis data, and Lagrangian backward trajectory calculations. We show that extreme DTDT changes in the extratropics are generally associated with changes in air mass transport, in particular shifts from warm to cold air advection or vice versa, linked to regionally specific synoptic-scale circulation anomalies (ridge or through patterns). These dominant effects of

advection are modulated by changes in adiabatic and diabatic processes in the transported air parcels, which tend to either amplify or dampen DTDT decreases (cold events) and increases (warm events) depending on the region and season. In contrast, DTDT extremes during December-February in the tropics are controlled by local processes rather than changes in advection. For instance, the most significant DTDT decreases are associated with a shift from less cloudy to more cloudy conditions, highlighting the crucial role of solar radiative heating. The mechanistic insights into extreme DTDT changes

obtained in this study can help improve the prediction of such events and anticipate future changes in their occurrence frequency and intensity, which will be investigated in part II of this study.

## 1.   Introduction

Day-to-day temperature (DTDT) variations, here represented by temperature differences between consecutive days, can have significant implications across various sectors, including economic, ecology, and human health (Gough, 2008; Zhou et al.,

2020; Kotz et al., 2021; Hovdahl, 2022). A change in DTDT variability is linked to increased mortality rates, with the impact varying across geographic regions such as northern latitudes, tropics, and southern latitudes (Chan et al., 2012; Hovdahl, 2022; Sarmiento, 2023; Wu et al., 2022a). A recent study further reveals that variations in temperature variability can impact heat-related mortality by regionally up to more than 7% of total deaths (Wu et al., 2022b). High DTDT variability has a negative effect on economic activity (Linsenmeier, 2023), with economic losses exhibiting regional disparities, as a 1°C

increase in DTDT variability results in an increase of 3-5% in vulnerability in mid and high-latitude areas but more than 10%





in low-latitude and coastal regions (Kotz et al., 2021). These previous studies highlight the critical need for comprehensive studies investigating DTDT variability and the underlying atmospheric processes in the present climate.

With regard to long-term trends in DTDT variability, the early study by Karl et al. (1995) observed a decrease at mid-to-high

latitude stations, particularly during the summer season, while no significant trend was detected in Australia. Recently, Xu et al. (2020) extended these results by demonstrating a substantial increase in summer DTDT variability across diverse regions such as the Arctic coast, South China, and Australia but decreasing winter DTDT variability in mid to high latitudes. These trends align with the comprehensive analysis by Krauskopf and Huth (2024) and Wan et al. (2021), attributing spatiotemporal fluctuations in global variability, except during boreal summer, to anthropogenic influences. Although most

prior investigations have predominantly focused on analyzing trends in DTDT variability, it is equally important to understand DTDT extremes. The concept of a DTDT extreme events, characterized by daily average temperature changes larger than ±10°C, has been introduced in a recent study only focusing on mid-high latitudes (Zhou et al., 2020). Since higher DTDT variability is observed in the northern mid-high latitudes, spanning regions such as North Asia, the United States, and Europe (Gough, 2008; Xu et al., 2020), we can also expect more pronounced DTDT extremes in these regions

compared to lower latitudes (Figures 3a-d), which is consistent with the findings of (Zhou et al., 2020). Here, to account for such regional differences and investigate DTDT extremes globally, we utilize percentile-based thresholds, as widely used for identifying temperature extremes (Bieli et al., 2015; Pfahl, 2014; Nygård et al., 2023).

The relationship between temperature extremes (extremely high or low temperature, not DTDT extremes) and distinct

circulation patterns at global and regional scales has been thoroughly investigated (Horton et al., 2015; Adams et al., 2021; Nygård et al., 2023; Pfahl, 2014). For example, in Europe, extreme warm events in summer often coincide with blocking anticyclones or subtropical ridges, while winter cold extremes are linked to North Atlantic blocking, facilitating the intrusion of cold air masses (Nygård et al., 2023; Kautz et al., 2022; Pfahl and Wernli, 2012; Sillmann et al., 2011). Similarly, warm extremes in North America correlate with anticyclonic circulation and ridges, whereas extreme cold is linked to troughs and

advection of continental air masses (Adams et al., 2021; Wang et al., 2019). Particular insights into the associated physical processes have been obtained from Lagrangian studies based on air parcel trajectory analyses, which have quantified the contributions of advection, adiabatic, and diabatic warming/cooling to temperature extremes (Schumacher et al., 2019; Bieli et al., 2015; Röthlisberger and Papritz, 2023b, a; Zschenderlein et al., 2019; Hartig et al., 2023). All these studies highlight the significance of specific circulation anomalies for causing extreme weather in mid-to-high latitudes. In contrast, tropical

regions typically exhibit weaker temperature advection. However, an accelerated warming of extreme temperatures across tropical land has been observed recently (Byrne, 2021). These changes can be linked to shifts in wind patterns, precipitation, radiation, cloud cover, and surface fluxes (Gough, 2008; Matuszko et al., 2004; Sun and Mahrt, 1995; Dirmeyer et al., 2022). These previous studies have established groundwork for understanding the origins of temperature extremes, providing a fundamental basis also for deeper investigations into DTDT extremes globally.




This paper investigates in detail the influence of altering air mass properties within the large-scale atmospheric circulation on the variability of DTDT extremes. We use observations and reanalysis data to quantify the magnitude of DTDT extremes worldwide. To study the underlying processes, we apply a composite approach and perform a Lagrangian analysis, calculating backward trajectories of surface air masses from selected locations on the two days involved in DTDT extremes.

The contributions of temperature advection, adiabatic, and diabatic warming are then quantified following previous studies of temperature extremes (Bieli et al., 2015; Röthlisberger and Papritz, 2023a; Santos et al., 2015; Röthlisberger and Papritz, 2023b; Nygård et al., 2023). We aim to address the following research questions: (1) Which (changes in) atmospheric circulation patterns occur on consecutive days associated with DTDT extremes? (2) Which physical processes contribute to the occurrences of DTDT extremes?

**2. Data and Method**

**2.1 Observations**

HadGHCND is a gridded daily temperature data set that includes near-surface minimum and maximum temperature data from 1950 to the present, developed by the National Centers for Environmental Information. It is based on meticulously quality-controlled observations from ~27,000 global stations within the Global Historical Climatology Network daily
database (Caesar et al., 2006). The data have been interpolated on a 2.5° latitude by 3.75° longitude grid, employing an angular distance weighting scheme for spatial consistency. Specifically designed to analyze climate extremes and climate model evaluation, the dataset offers temperature anomalies (based on a 1961-1990 climatology) and actual temperatures separately. Given its comprehensive nature and rigorous quality control, the HadGHCND data provides a robust foundation for global climatic studies (Xu et al., 2020; Wan et al., 2021). For the current study, we have used global daily HadGHCND
(minimum (TN) and maximum (TX)) temperature from 1980 to 2014 and then calculated the daily mean temperature.

**2.2 Reanalysis data**

ERA5 is the fifth-generation European Centre for Medium-Range Weather Forecasts (ECMWF) global reanalysis product, providing climate and weather data for the past eight decades. It was developed by 4D-Var data assimilation in cycle CY41R2 of the ECMWF Integrated Forecasting System (Hersbach et al., 2020). It is an updated version of the widely used
ERA-Interim reanalysis, employing a newer version of the ECMWF Earth system model with 137 hybrid sigma/pressure levels in the vertical up to 0.01 hPa (Hersbach, 2019). It provides hourly estimates of several atmospheric, ocean-wave, and land-surface quantities on a regular latitude-longitude grid of 0.25 degrees. Recently, ERA5 data has been used for different climate and weather-related global and regional studies (Böker et al., 2023; Simmons, 2022), and some have recommended it as the best alternative for regions with sparse observational coverage (Sharma et al., 2020; Sheridan et al., 2020). We use
ERA5 data from 1980 to 2020, encompassing global daily mean 2m-surface air temperature (calculated from hourly





temperature), total precipitation, and several three-dimensional atmospheric fields (temperature, horizontal and vertical winds, geopotential). The spatial resolution for all analyses is $0.25° \times 0.25°$, except for the trajectory calculations, for which input data at a horizontal resolution of $0.5° \times 0.5°$ are employed. The temporal resolution of the near-surface temperature and composite analysis is daily, while the input data for the trajectories have an hourly resolution.

**2.3 DTDT variability and extremes**

This study defines the DTDT change, $\delta_T$, as the difference in daily mean near-surface air temperature between the previous (t-1) and events day (t), as shown in Eq. (1).

$$\delta_T = (T_t - T_{t-1}) \qquad (1)$$

$$\mu_T = \frac{1}{n} \sum_{t=1}^{n} (T_t - T_{t-1}) = T_n - T_0 \qquad (2)$$

$$\sigma_{DTDT}{}^2 = \frac{1}{n} \sum_{t=1}^{n} (T_t - T_{t-1})^2 \qquad (3)$$

$$= \frac{1}{n} \sum_{t=1}^{n} ((T_t - \mu_T) - (T_{t-1} - \mu_T))^2$$

$$= \frac{1}{n} \sum_{t=1}^{n} ((T_t - \mu_T)^2 + (T_{t-1} - \mu_T)^2 - 2(T_t - \mu_T)(T_{t-1} - \mu_T))$$

$$\approx 2\sigma_T^2 - 2COV(T_t, T_{t-1})$$

$$\sigma_{DTDT} = \sigma_T \sqrt{2(1 - ACORR(T_t, T_{t-1}))} \qquad (4)$$

$$\sigma_{DTDT} = \sigma_T \sqrt{2(1 - r_{1,T})} \qquad (5)$$

Given that $T_t$ and $T_{t-1}$ are the near-surface air temperature on the event and previous day. The average over these daily changes, $\mu_T$, only reflects the difference between the daily temperature at the beginning and end of the time series (Eq. 2), we use their standard deviation $\sigma_{DTDT}$ to quantify typical DTDT changes. This standard deviation can be expressed as a function of the usual standard deviation $\sigma_T$ and the lag-1 autocorrelation $r_{1,T}$ of daily mean temperature, as shown in Eq. (5).

In addition, extreme DTDT changes are studied using the percentile method. Cold and warm extreme DTDT change events at each grid point are identified using the 5th and 95th percentiles of the DTDT change as thresholds. The analyses focus on the two main seasons: December-February (DJF) and July-August (JJA). Accordingly, 184 events in DJF and 188 events in JJA are selected at each location.

**2.4 Trajectory setup**

The Lagrangian analysis tool (LAGRANTO) introduced by Sprenger and Wernli (2015) is employed to calculate backward trajectories of near-surface air masses on the days involved in an extreme  DTDT change events. The trajectories are initialized at 18 UTC on both days and at 10, 30, 50, and 100 hPa above the surface at the corresponding grid points. Various



variables of interest, including latitude, longitude, pressure, temperature, and potential temperature, are interpolated along the trajectory paths and saved in 1-hour intervals. The initialization time of the trajectories in the following is referred to as t= 0.

## 2.5 Lagrangian temperature decomposition

To better understand the underlying mechanisms, our analysis focuses on four specific locations: two grid points in the Northern Hemisphere mid-latitudes (North America and Europe), one in tropical South America, and another on the southern coast of Australia. At these locations, we apply a novel Lagrangian temperature decomposition method to quantify contributions of advection, adiabatic, and diabatic processes to DTDT extremes - similar to approaches used for near-surface hot and cold extremes (Röthlisberger and Papritz, 2023b, a). This decomposition is applied to backward trajectories initiated on the two days involved in an extreme DTDT change events. A detailed derivation of the diagnostic is presented in the appendix. In summary, the Lagrangian decomposition of DTDT changes, as approximated by the trajectories, and here evaluated on a 3-day Lagrangian time scale, is given in Eq. (6).

$$\delta_T^0 \approx \delta_{\overline{T}}^{-3d} + \delta_{\overline{T}}^{adi} + \delta_{\overline{T}}^{dia} + res \qquad (6)$$

It is decomposed into three contributing factors: the mean temperature difference at the origin of the air parcels three days before initialization, which indicates the contribution of advection ($\delta_{\overline{T}}^{-3d}$), a contribution of mean adiabatic compression or expansion resulting from vertical descent or ascent, respectively ($\delta_{\overline{T}}^{adi}$), and a contribution of mean diabatic heating or cooling from processes such as latent heating in clouds, radiation, and surface fluxes ($\delta_{\overline{T}}^{dia}$). The final term is the residuum (res), resulting from numerical inaccuracies in derivative calculations.

## 3. Result

### 3.1 DTDT variations in DJF and JJA

Both the HadGHCND and ERA5 datasets reveal that the magnitude of $\sigma_{DTDT}$ changes is larger in the extratropics than in the tropics during DJF and JJA (Figures 1a-b and 2a-b, see zonal average plot as well). Notably, $\sigma_{DTDT}$ variability is larger during DJF than in JJA. This difference is particularly evident in North America, the Northern Eurasian continent, and southern Australia. In DJF, the $\sigma_{DTDT}$ variability is above 3°C in many regions (Figures 1a-b), compared to 1-4°C during JJA (Figures 2a-b). However, $\sigma_{DTDT}$ variations remain consistently around 1-2°C in the tropics and 1-3°C over higher-latitude land regions in the southern hemisphere (except southern Australia) during both seasons. Furthermore, the ERA5 dataset indicates a considerably larger magnitude of $\sigma_{DTDT}$ changes than HadGHCND, primarily in northern hemisphere high





latitudes and southern Australia. This may be related to the lack of adequate station coverage in these regions (Fig. S1 of Wan et al. (2021)).

Since the magnitude of $\sigma_{DTDT}$ changes can be expressed as a function of the standard deviation $\sigma_{T,}$ and lag-1 autocorrelation $r_{1,T}$ of daily mean temperature (see again Eq. 5), Figures 1 and 2 also show these related quantities and their respective zonal averages. While HadGHCND and ERA5 mostly agree on the spatial pattern of $\sigma_T$, there is a slight overestimation of the magnitude of $\sigma_T$ in ERA5 regionally during DJF (Figures 1c-d). This overestimation is more widespread and larger in JJA (Figures 2c-d). The spatial distribution of $\sigma_T$ follows a pattern consistent with $\sigma_{DTDT}$, with generally higher variability in northern hemisphere mid-high latitudes (5-10 °C in DJF and 2-6°C in JJA) and southern hemisphere extratropics (2-6°C in

both seasons). In comparison, the tropics exhibit smaller variability (1-3°C in DJF and JJA) of $\sigma_T$.

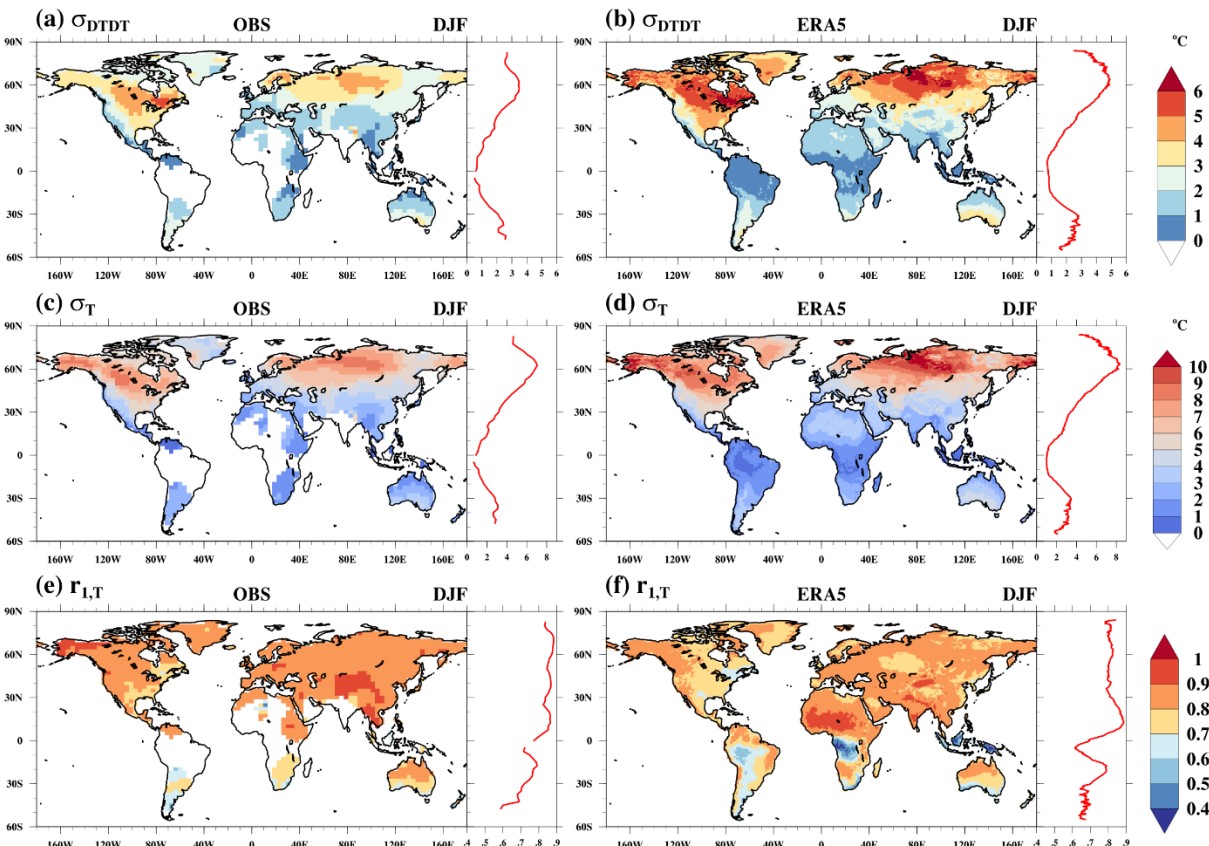

**Figure 1. (a, b) Standard deviation of DTDT variations ($\sigma_{DTDT}$, °C), (c, d) standard deviation of daily mean temperature ($\sigma_T$, °C), and (e, f) lag-1 autocorrelation of daily mean temperature ($r_{1, T}$) in December-February (DJF) derived from the HadGHCND (1st column) and ERA5 (2nd column) datasets. The zonal average of the $\sigma_{DTDT}$, $\sigma_T$, and $r_{1,T}$ are attached to each plot.**





In HadGHCND, the autocorrelation is spatially rather homogeneous, while there are more pronounced spatial variations with generally lower correlations in the ERA5 datasets (Figures 1e-f and 2e-f). Autocorrelation values are typically below 0.8 (and locally below 0.6) in the deep tropics, eastern North America, the southern hemisphere land regions south of
approximately 30°S, and the eastern half of the Asian continent in JJA, and above 0.8 in other regions.

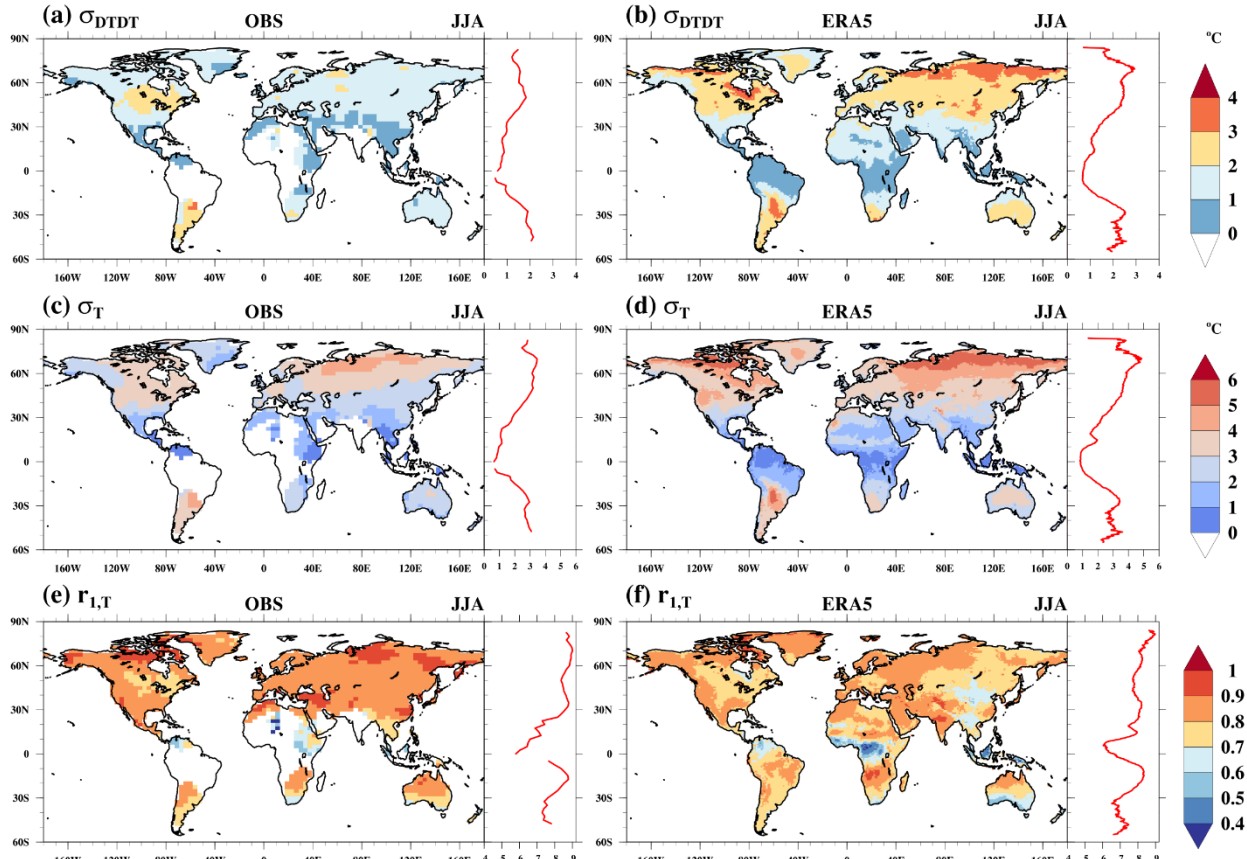

**Figure 2. (a, b) Standard deviation of DTDT variations ($\sigma_{DTDT}$, °C), (c, d) standard deviation of daily mean temperature ($\sigma_T$, °C), and (e, f) lag-1 autocorrelation of daily mean temperature ($r_{1,T}$) in June-August (JJA) derived from the HadGHCND (1st column)**
**and ERA5 (2nd column) datasets. The zonal average of the $\sigma_{DTDT}$, $\sigma_T$, and $r_{1,T}$ are attached to each plot.**

Comparing the global patterns of $\sigma_{DTDT}$, $\sigma_T$, and $r_{1,T}$ shows that the spatial variability of $\sigma_{DTDT}$ is mostly determined by variations in $\sigma_T$: A high standard deviation $\sigma_T$ leads to typically large $\sigma_{DTDT}$ changes in higher latitudes, despite a relative high $r_{1,T}$ (see again Eq. 5), while in the tropics, $\sigma_{DTDT}$ is smaller associated with lower $\sigma_T$, despite lower $r_{1,T}$. Nevertheless, $r_{1,T}$
can affect the spatial pattern of $\sigma_{DTDT}$ regionally. For instance, the north-south gradient of $r_{1,T}$ over Australia in JJA leads to larger $\sigma_{DTDT}$ changes in the south despite relatively homogeneous $\sigma_T$. With regard to the differences between HadGHCND and ERA5, both a higher $\sigma_T$ and lower $r_{1,T}$ in ERA5 contribute to the typically larger $\sigma_{DTDT}$ changes in the reanalysis data.





## 3.2 DTDT extremes

To investigate extreme DTDT changes, we use the 5th and 95th percentiles as thresholds at each grid point, as illustrated in
Figures 3a-d. The spatial patterns of extreme DTDT, both warm and cold, closely resemble those of $\sigma_{DTDT}$ and $\sigma_T$, with higher values in the extratropics and lower values in the tropics (Figures 1, 2, and 3). This similarity suggests that regions with greater DTDT variability are more prone to extreme DTDT events. Furthermore, extreme DTDT events are more intense (i.e., exhibit a larger magnitude) during DJF than JJA, particularly in the Northern Hemisphere. In contrast, the tropics display almost equal magnitudes of extreme DTDT in both seasons. The following section describes the atmospheric
circulation and physical processes (according to Eq. 6) related to extreme DTDT events at selected global locations.

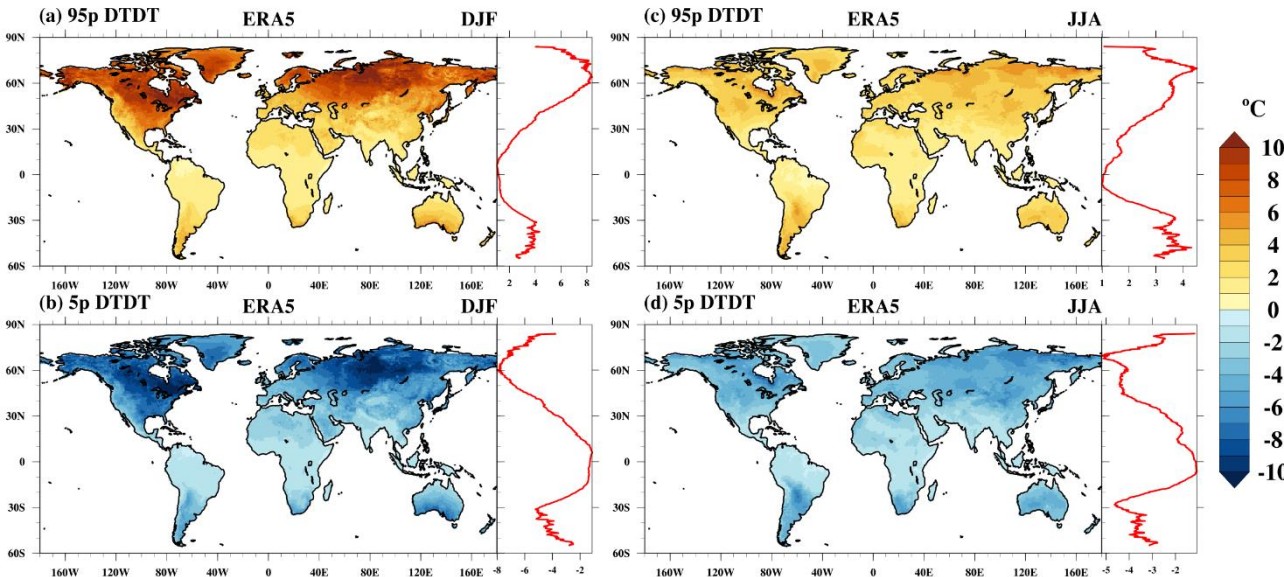

**Figure 3. The (a, c) 95th percentile (95p) and (b, d) 5th percentile (5p) of DTDT variations during (a, b) December-February (DJF) and (c, d) July-August (JJA) as derived from the ERA5 dataset. The zonal average of the 95p and 5p of DTDT changes are**
**attached to each plot.**

### 3.2.1 Mid-latitudes: North America

To explore the mechanism behind extreme DTDT changes in the mid-latitudes, we focus on a specific grid point in North America (52°N and 85°W), which is also representative of other North American regions (compare results with Fig. S2).

The synoptic-scale circulation patterns associated with DJF warm events (exceeding the 95th percentile of DTDT changes) on consecutive days are first analyzed with the help of composites, as shown in Figures 4a-b. On the day preceding the events (t-1), a distinct near surface-temperature dipole is observed across northern North America, with higher temperatures in the west (≥-10°C) and lower temperatures (≤-15°C) in the east and the vicinity of the selected location (Figure 4a). This





temperature pattern is influenced by a ridge over the western part facilitating warm, southerly airflow around its western

flank and a trough over the eastern part of the continent associated with northerly flow and cold air advection. This synoptic pattern aligns with the distribution of backward trajectories, indicating the advection of cold air masses from the Arctic region (Figure 5a). Over the 3d leading up to this preceding day, cold air masses (mean temperature of -21.5°C at -3d) experience a gradual temperature increase (of 5.7 °C), with significant adiabatic warming (8.3°C in the mean) due to a distinct 100hPa mean descent (Figures 5e-f). Limited diabatic cooling, likely due to longwave radiation, is indicated by a

reduction in θ (by -2.6°C in the mean), constraining the temperature increase (Figure 5g). Note that the residual is below 1°C (all locations), attributable to numerical inaccuracies in the computation of derivatives, which is negligible. Accordingly, the temperature at t-1 is mainly determined by cold air advection, mitigated by adiabatic warming, with diabatic processes contributing to some additional cooling.

On the day of the DJF warm events (t), the near-surface temperature reaches ≥-10°C on average, marking a notable increase compared to the t-1 (Figure 4b). This rise is attributed to an eastward extension of the ridge, displacing the preceding trough further to the east, which leads to a southwesterly advection at the selected location. The air parcel density shifts substantially southward compared to t-1, with the largest density southwest of the chosen location (40-50°N, Figure 5b). This shift is associated with a substantially higher temperature 3d before arrival (-13.4°C in the mean) compared to the air parcels

at t-1. The air parcels experience a vertical descent of 76hPa in the mean (Figure 5e), leading to adiabatic warming (6.7°C) and a general temperature increase in the 3d leading up to the events (Figure 5f). The net diabatic heating contribution is close to zero (0.4°C over 3d) but becomes more prominent in the last hours before the events, reaching 1.2°C in 24h (Figure 5g). Similar to t-1, the initially colder air masses are thus warmed on their way to the target location. Evaluating the differences in the physical processes between the two successive days (Eq. 6) reveals that changes in advection, associated

with the shift in the air mass origin and initial temperature, are the main cause for the local DTDT warming at the selected location, contributing 8.1°C on average (Figure 5k). Reduced adiabatic warming (-1.6°C) due to a less pronounced descent on the day of the events counteracts the overall temperature increase, while increased diabatic heating (3°C) provides another positive contribution. This diabatic contribution is associated with the largest events-to-events variability (boxes and whiskers in Figure. 5k).


Warm events in JJA resemble those of DJF, including their atmospheric circulation patterns (not shown) and air parcel density distribution (Figs. S1a-b). The air parcels also experience similar adiabatic warming before arriving at the target location as in DJF, whereas the diabatic changes have a larger daily cycle and a stronger positive contribution on the last day (Figs. S1e-g). The warming, as in DJF, is mainly due to the change in the air parcels' initial temperature (0.9°C at t-1 and

6.4°C at t) and thus changes in advection (Fig. S1k). Mean contributions from adiabatic warming (1.9°C) and diabatic heating (-0.8°C) are smaller magnitude and flip signs compared to DJF.



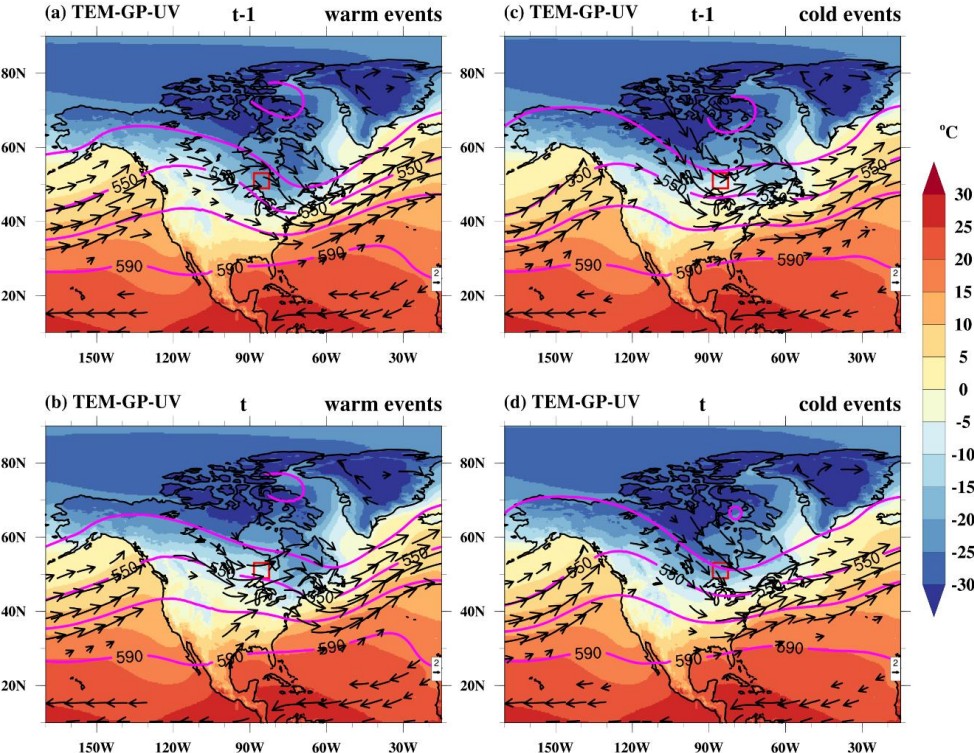

**Figure 4. Composite of near-surface temperature (TEM, °C, color shading), wind at 850 hPa (UV, m/s, vectors), and Geopotential**
**height at 500 hPa (GP, gpm, magenta contours) on the (a, c) previous day (t-1) and (b, d) event day (t ) of the warm (a,b) and cold**
**events (c,d) during December-February (DJF) at a selected grid point in North America (red box). Note that wind vectors ≥5m/s**
**are plotted.**

On the days preceding DJF cold events (t-1), there is a pronounced meridional temperature gradient in the region of the
selected location, with near-surface temperature over the northern Arctic land region below -15°C, while milder
temperatures (≥-10°C) prevail at the grid point and to its south (Figure 4c). This thermal gradient is linked to a trough over
the Arctic juxtaposed with a ridge structure to its west. Westerly winds facilitate the transport of relatively warm air masses
from the west towards the location (Figure 5c). The transported air parcels, with a mean temperature of -15°C (at -3d),
undergo a gradual descent, experiencing a modest temperature increase of about 3.7°C in 3d, attributed to the combined
effects of adiabatic (2.4°C) and diabatic warming (1.3°C), as shown in Figures 5h-j. While the descent and adiabatic
warming dominate between -3d and -2d, the last day before arrival is characterized by diabatic warming (θ increase by 2°C,
Figure 5j). As for the t-1 events, the near-surface temperature at the selected location is thus mainly determined by
advection, with some additional warming due to diabatic and adiabatic processes.



On the day of the events (t), a southward shift of the Arctic trough is associated with a turn of the wind towards the north-northwest at the selected location (Figure 4d). This corresponds to the arrival of northerly air parcels with an initially extremely low temperature of -26.2°C in the mean (Figure 5d). These air parcels undergo a subsequent temperature increase of 3.2°C in 3d due to adiabatic warming (4.3°C) partly offset by radiative cooling (-1.1°C, Figures 5h-j). Accordingly, a shift to strong cold advection, associated with a mean temperature difference of -11.2°C between the air parcels at -3d, is the

predominant contributor to the DTDT cold events (Figure 5l). This advective effect is counteracted by moderately increased adiabatic warming (1.9°C) and complemented by amplified diabatic cooling (-2.4°C), collectively shaping the DTDT change.

In JJA, the DTDT cold events are driven by very similar processes as the cold events during DJF, with a shift from westerly

to northerly transport (Figs. S1c-d) leading to a mean temperature drop of -8.6°C due to advection, while changes in adiabatic and diabatic warming are very small (Fig. S1l).





**Figure 5.** The spatial distribution of trajectories initiated on the previous day (t-1) and event day (t) for both December-February
(DJF) warm and cold events over North America. In the top row, the color-shading illustrates the air parcel trajectory density (%)
based on the position between -5d and 0d. The magenta and yellow contours represent 0.5% particle density fields at -3d and -1d,





**respectively. The red box shows the selected grid point over North America. The Lagrangian evolution of distinct physical parameters (pressure, temperature, potential temperature) along the air parcel trajectories for both warm (2ⁿᵈ row) and cold events (3ʳᵈ row) is presented in panels e-j. Panels k and l show the contribution of the different physical processes to the genesis of**

**DTDT extremes according to Eq. (6), which refers to a 3d-time scale. The residuum is attributable to numerical inaccuracies in the computation of derivatives in Eq. (6). The box spans the 25th and 75th percentile of the data; the black dot inside the box gives the mean of the related quantities, and 1.5 times the interquartile range is indicated by the whiskers.**

### 3.2.2 Mid-latitudes: Europe

Another mid-latitude location where we investigate the mechanism driving extreme DTDT variations is a grid point over

Europe (50°N and 10°E). This section focuses on the JJA season since the DJF events, which are only briefly discussed, exhibit similarities with the location in North America studied in section 3.2.1.

During the preceding day of the JJA warm events (t-1), the composite geopotential height pattern features a weak trough over the eastern North Atlantic and a developing ridge over Central Europe (Figure 6a). In this situation, the northern part of

Central Europe, including the selected location, is under the influence of westerlies transporting cool, maritime air masses towards the continent (cf. Figure 7a), associated with mean temperatures below 18°C. On the contrary, Spain and western France are already affected by southwesterly winds associated with the approaching ridge, leading to higher temperatures (≥21°C) there. The mean initial temperature of the tracked air parcels 3d before arriving at the selected location is 5.4°C, and they subsequently undergo a temperature increase of about 10.1°C in 3d (Figure 7f). This increase is partly due to adiabatic

warming, corresponding to a mean temperature rise of 5.7°C, which is linked to a mean pressure increase of 62 hPa. It is worth noting that this subsidence is higher between 3d and 1d before arrival and subsequently slows down when the air parcels get close to the surface (Figure 7e). Diabatic warming, likely resulting from surface fluxes, increases the air temperature by 4.4°C in the mean, mainly on the last day before arrival. Therefore, all three processes—advection, adiabatic, and diabatic warming—contribute to determining the near-surface temperature at t-1 (Figure 7k).


On the day of the events (t), the trough-ridge pattern typically shifts eastward, such that Central Europe also gets under the influence of the southwesterly flow ahead of the trough (on the western flank of the ridge) and the near-surface temperature rises above 18°C (Figure 6b). This warming is associated with the arrival of air masses from western continental Europe (Figure 7b). These air parcels have a mean initial temperature of 9.3°C at -3d, which is substantially warmer than the air

masses arriving at t-1, with a subsequent temperature increase of 11.4°C in 3d (Figure 7f). This warming is attributed to adiabatic warming (4.5°C in the mean, mean descent of 47hPa) and strong diabatic heating (6.9°C in the mean). Comparing the contribution of processes between the two days shows that warm advection (3.9°C, Figure 7k) is the predominant factor driving the DTDT increase. Further warming is facilitated by increased diabatic processes (2.5°C in the mean). Adiabatic warming (-1.2°C), on average, has a small negative contribution (descent is larger at t-1 compared to t), albeit with large

variability between events (Figure 7k).



DJF warm events are dominated by a ridge pattern over western Europe, which weakens on the day of the events, associated with a more zonally oriented flow bringing a larger fraction of warmer maritime air masses to the selected location, which is in contrast to the mainly continental air parcel origin at t-1 (Figs. S3a-b and S4a-b). This change in origin and the associated warm advection and adiabatic warming are the main causes of the DTDT increase, whereas diabatic warming does not differ greatly between the two days (Fig. S4).

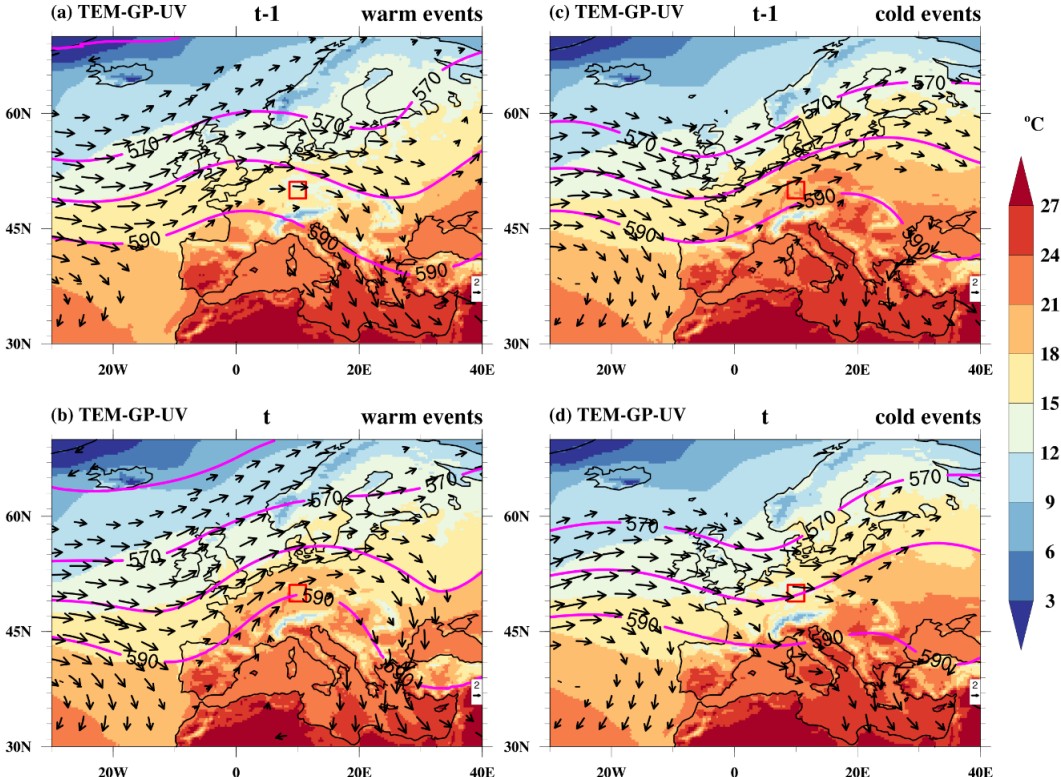

**Figure 6. Composite of near-surface temperature (TEM, °C, color shading), wind at 850 hPa (UV, m/s, vectors), and Geopotential height at 500 hPa (GP, gpm, magenta contours) on the (a, c) previous day (t-1) and (b, d) event day (t ) of the warm (a,b) and cold events (c,d) during  June-August (JJA) at a selected grid point in Europe (red box). Note that wind vectors ≥5m/s are plotted.**

During the day preceding JJA cold events (t-1), relatively high near-surface temperatures (≥18°C) in Central Europe are associated with the southwesterly flow and warm air advection linked to a trough over the British Isles and a ridge downstream over Scandinavia (Figures 6c, 7c). The air parcels are already relatively warm 3d before they arrive at the selected location (mean temperature of 13.1°C, Figure 7i) and are further heated by 5.9°C in 3d. Both adiabatic warming during their descent (3.7°C, Figures 7h-i) and diabatic heating (2.2°C, Figure 7j) contribute to this temperature increase.



On the day of the events (t), Central European temperature decreases substantially to values below 18°C, linked to a weakening of the upstream trough and eastward propagation of the ridge, leading to more zonal flow conditions (Figure 6c).

Northwesterly winds carry cold, maritime airmasses toward the target location, with an initial temperature of 5.4°C (Figure 7d and Figure 7i). These air parcels are further warmed by 7.3°C due to strong adiabatic warming  (5.7°C, mean descent of 60hPa, Figure 7h) and some diabatic warming (1.6°C, Figure 7j) in 3d. Comparing the process contributions on the two days shows that DTDT cold events occur predominantly due to a shift from warm to cold air advection (mean temperature change of -7.7°C, Figure 7l). This advective effect is partly balanced by an increase in adiabatic warming (2°C). The reduced

diabatic warming is small but possesses substantial events-to-events variability (Figure 7l).

DJF cold events are triggered by a shift from westerly to northerly winds ahead of a developing ridge over the eastern North Atlantic (Figs. S3c-d and Figs. S4c-d). Air masses arriving at the day of the events (t) have a much lower temperature at their origin (-3d) compared t-1 due to this northward shift in the source location (Figs. S4c-d) and a typically higher altitude (Fig.

S4h). This leads to a pronounced contribution of advection to the DTDT decrease, which is, however, partly compensated by increased subsidence and adiabatic warming on the day of the events and a slight increase in diabatic heating (Figs. S4h,i,j,l).




**Figure 7.** The spatial distribution of trajectories initiated on the previous day (t-1) and event day (t) for both June-August (JJA) warm and cold events over Europe. In the top row, the color-shading illustrates the air parcel trajectory density (%) based on the position between -5d and 0d. The magenta and yellow contours represent 0.5% particle density fields at -3d and -1d, respectively. The red box shows the selected grid point over Europe. The Lagrangian evolution of distinct physical parameters (pressure, temperature, potential temperature) along the air parcel trajectories for both warm (2nd row) and cold events (3rd row) is





**presented in panels e-j. Panels k and l show the contribution of the different physical processes to the genesis of DTDT extremes according to Eq. (6), which refers to a 3d-time scale. The residuum is attributable to numerical inaccuracies in the computation of derivatives in Eq. (6). The box spans the 25th and 75th percentile of the data; the black dot inside the box gives the mean of the related quantities, and 1.5 times the interquartile range gives the whiskers.**

### 3.2.3 Tropics: South America

To systematically investigate the mechanism of extreme DTDT changes in the deep tropics during DJF and JJA, we select a specific location in South America (56°W and 13°S) and also compare results with South Africa (24°E and 13°S, Figs. S6
and S7), where physical processes appear to be similar.

During JJA, DTDT extremes over tropical South America are associated with distinct patterns, particularly in the wind field (Figs. S5a-d). At the day preceding the events (t-1), the selected location lies in the region of a strong horizontal temperature gradient, with higher temperatures (≥22°C) northeast and lower temperatures (≤20°C) southwest of it. The latter may be
associated with extratropical influences, mainly through a trough over Argentina and the South Atlantic (Fig. S5a). Air parcels arriving on this day originate (at -3d) mainly from the south (Figure 8a), with a mean initial air temperature of 9.7°C that gradually increases due to adiabatic warming (4°C, descent of 41 hPa) and strong diabatic warming (9.9°C, Figures 8e-g).

On the day of the events (t), the trough weakens, and winds turn easterly, which is associated with a larger fraction of air parcels originating from the east (Fig. S5b and Figure 8b). These air parcels are initially warmer (12.7°C on average) than at t-1 (Figure 8f) and again experience an average temperature increase (by 13.9°C), influenced by adiabatic warming (5.6°C, 58hPa descent) and strong diabatic warming (8.3°C, Figures 8e-f). When examining the physical processes across consecutive days, the DTDT warm events can be attributed to a combination of factors, including warm air advection,
enhanced adiabatic heating, and reduced diabatic heating (Figure 8k). Initially, the warm advected air masses experience heating during their descent; this temperature increase is mitigated by a less pronounced diabatic warming on the events day (Figures 8e-g). Therefore, the primary contribution to warming comes from advection and some additional adiabatic warming.





**Figure 8. The spatial distribution of trajectories initiated on the previous day (t-1) and event day (t) for both June-August (JJA) warm and cold events over South America. In the top row, the color-shading illustrates the air parcel trajectory density (%) based on the position between -5d and 0d. The magenta and yellow contours represent 0.5% particle density fields at -3d and -1d, respectively. The red box shows the selected grid point over South America. The Lagrangian evolution of distinct physical**



**parameters (pressure, temperature, potential temperature) along the air parcel trajectories for both warm (2nd row) and cold**
**events (3rd row) is presented in panels e-j. Panels k and l show the contribution of the different physical processes to the genesis of DTDT extremes according to Eq. (6), which refers to a 3d-time scale. The residuum is attributable to numerical inaccuracies in the computation of derivatives in Eq. (6). The box spans the 25th and 75th percentile of the data; the black dot inside the box gives the mean of the related quantities, and 1.5 times the interquartile range gives the whiskers.**

In contrast to JJA, DTDT extremes in DJF are not associated with specific circulation patterns. The origins of air parcels at -3d are clustered around the selected location, indicating that primarily local effects lead to the DTDT changes (Figures 9a-d). Negative contributions of advection and increased adiabatic warming cancel each other, such that DTDT changes are predominantly due to differences in diabatic heating on the last day before the air parcels arrive at the selected location (Figures 9e-l). To better understand these local, diabatic effects, we analyse composites of cloud cover and precipitation
based on ERA5 data (Figure 10a). For DTDT warm events, high cloud coverage (90-95%), along with substantial cumulative precipitation (8-10 mm/day), is observed at t-1 across the study region, resulting in reduced diabatic heating and colder temperatures (Figures 9f, g and 10a). In contrast, cloud cover (70-75%) and precipitation (2-4 mm/day) decrease on the day of the events, contributing to larger diabatic heating and higher temperatures (Figures 9f, g, k, and 10b). Thus, the DTDT change during warm events is linked to a transition from primarily cloudy and wet to less cloudy and drier conditions.
This indicates an essential role of albedo changes and solar radiative heating in triggering the temperature increase.

The day before JJA cold events (t-1) is characterized by high near-surface temperatures (≥24°C) at the selected location and a temperature gradient to the south (Fig. S5c). Easterly winds prevail over the study area, bringing in initially warm air parcels (17.3°C mean temperature at -3d) that undergo an additional temperature rise (7.7°C) due to adiabatic (3.3°C) and
diabatic heating (4.4°C). In contrast, on the day of the events, colder air masses are transported to the selected location by southerly winds upstream and equatorward of a subtropical trough over south-east South America and the South Atlantic (Figure 8d and Fig. S5d), again pointing to a potential role of extratropical-tropical interactions. The air parcels originally had lower temperatures (11.4°C) than at t-1. They undergo a temperature increase (10.8°C) due to adiabatic warming (4.4°C) and diabatic heating (6.4°C, Figures 8h-j). Comparing the two days indicates that JJA cold events are driven by cold air
advection, which is partly counterbalanced by increased adiabatic and diabatic warming (Figure 8l).



**Figure 9.** The spatial distribution of trajectories initiated on the previous day (t-1) and event day (t) for both December-February (DJF) warm and cold events over South America. In the top row, the color-shading illustrates the air parcel trajectory density (%) based on the position between -5d and 0d. The magenta and yellow contours represent 0.5% particle density fields at -3d and -1d, respectively. The red box shows the selected grid point over South America. The Lagrangian evolution of distinct physical parameters (pressure, temperature, potential temperature) along the air parcel trajectories for both warm (2nd row) and cold



events (3rd row) is presented in panels e-j. Panels k and l show the contribution of the different physical processes to the genesis of DTDT extremes according to Eq. (6), which refers to a 3d-time scale. The residuum is attributable to numerical inaccuracies in the computation of derivatives in Eq. (6). The box spans the 25th and 75th percentile of the data; the black dot inside the box gives the
mean of the related quantities, and 1.5 times the interquartile range gives the whiskers.

Similar to the DJF warm events, the DTDT change during DJF cold events is primarily driven by reduced diabatic heating on the last day before the air parcels arrive at the target location (Figures 9h-j), which can be attributed to variations in local conditions (Figures 10c-d). Again, like warm events, Figures 10c-d indicate that these variations are associated with changes

in cloud cover and precipitation: At t-1, relatively lower cloud cover (70% to 80%) is observed, coupled with lower total precipitation (4 to 8 mm/day) across the study area, ultimately resulting in higher temperatures (Figure 10c and Figure 9i). In contrast, on the day of the events, cloud cover (90-95%) and precipitation (14-20 mm/day) are higher, contributing to colder temperatures (Figure 10d and Figure 9i). Thus, DJF DTDT cold events are linked to a transition from less cloudy and drier to cloud-covered and wet conditions, again indicating a significant role of solar radiative heating (reduced diabatic heating, see

Figure 9l).

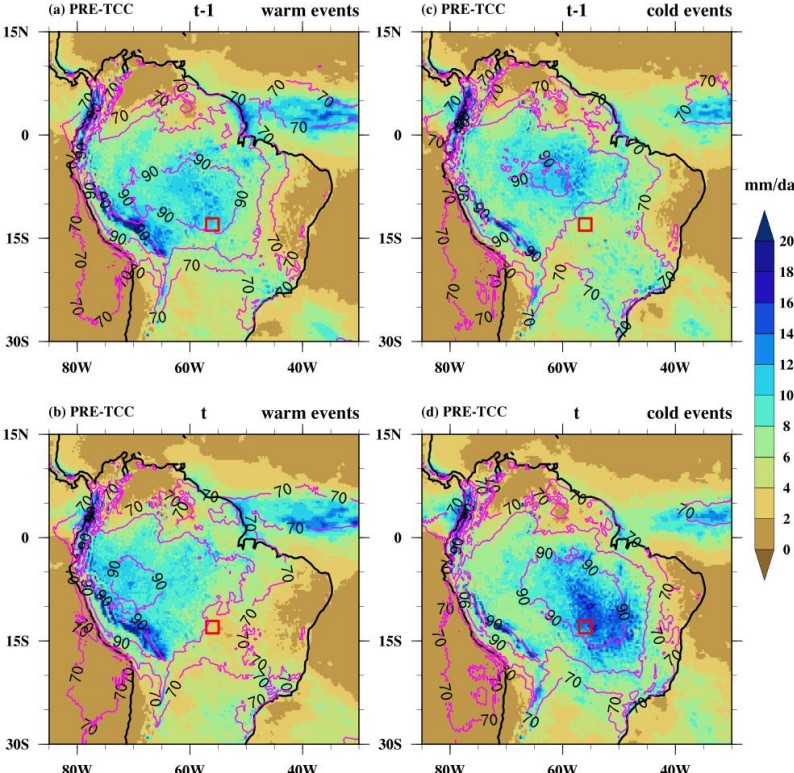

Figure 10. Composites of precipitation (PRE, mm/day, colour shading) and total cloud cover (TCC, %, magenta contours) on the (a, c) previous day (t-1) and (b, d) event day (t) of warm events (a,b) and cold events (c,d) during December-February (DJF) at a selected grid point in South America (red box).



### 3.2.4 Southern Hemisphere Subtropics: Australia

To systematically investigate the mechanism driving DTDT extremes over the subtropics in the southern hemisphere during DJF and JJA, we select a specific location in Australia (140°E and 37°S).

On the day preceding DJF warm events (t-1), the selected location lies in a region of weak winds in the center of a ridge and within a strong meridional temperature gradient, with higher temperatures over the continent to the north (≥24°C) and lower temperatures over the Southern Ocean (≤16°C, Figure 11a). The tracked air parcels, at -3d, are mostly located near the selected point or to its southwest (Figure 11a). Starting with relatively low temperatures (1.8°C on average), they undergo substantial subsidence (147 hPa) and adiabatic warming, resulting in a noteworthy temperature increase of 13.8°C (Figures 12e-f). In addition, they experience weak diabatic heating on the day before arrival (1.8°C, Figure 12g).

The temperature increase on the day of the events is associated with an intensification and eastward shift of a trough towards the Great Australian Bight and the resulting northerly airflow, bringing warm, continental air mass to the selected location (Figures 11b and 12b). These air parcels are slightly warmer (5.1°C) at -3d compared to t-1, also experience adiabatic warming during their vertical descent (9.4°C, mean descent of 98hPa), but are much more strongly affected by diabatic heating (9.9°C, Figures 12e-g). DJF warm events over the southern coast of Australia thus results from a shift from oceanic to continental air masses, resulting in amplified warm air advection (3.3°C) and, more importantly, increased diabatic heating (8.1°C), most likely due to surface fluxes over the warm continent, while reduced descent and adiabatic warming (-4.4°C) have a dampening effect (Figure 12l).

The atmospheric circulation and backward air parcel distribution during JJA warm events resemble those of DJF (Figs. S8a-b). However, most probably related to the weaker surface fluxes in Austral winter, diabatic heating does not contribute substantially to the temperature evolution along the trajectories. Consequently, the warm events are primarily due to warm air advection (5.9°C), with both the adiabatic (-2.2°C) and diabatic (-1.4°C) terms having a smaller damping effect (Fig. S8k).



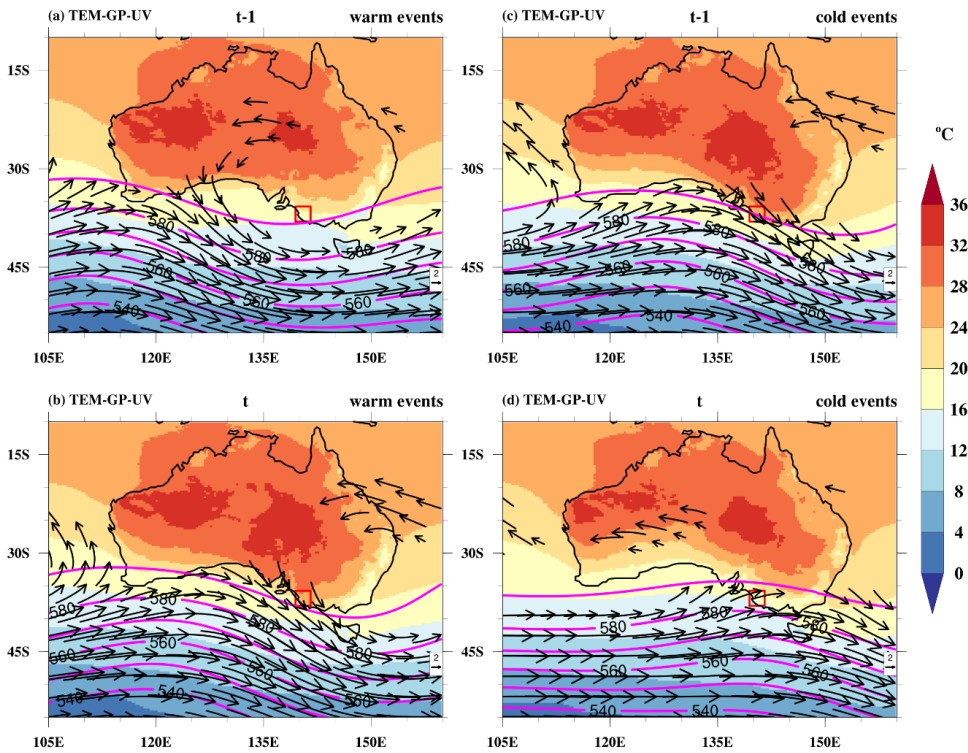


**Figure 11. Composite of near-surface temperature (TEM, °C, color shading), wind at 850 hPa (UV, m/s, vectors), and Geopotential height at 500 hPa (GP, gpm, magenta contours) on the (a, c) previous day (t-1) and (b, d) event day (t) of warm (a,b) and cold events (c,d) during December-February (DJF) at a selected grid point in Australia (red box). Note that wind vectors ≥5m/s are plotted.**


On the day preceding DJF cold events (t-1), a trough is located west of the selected grid point, associated with the advection of warm, continental air masses (Figures 11c and 12c). The air particles originate at a relatively high temperature (11.2°C) and undergo subsequent adiabatic warming (8.5°C, mean descent of 89 hPa), while diabatic heating is small (Figures 12h-j). Whereas on the day of the events, the trough has moved eastward, such that the selected location gets under the influence of

southwesterly flow on its western side (Figure 11d), leading to the advection of colder, oceanic air masses with a mean temperature of -1.8°C at -3d (Figures 12d-e). This shift to cold air advection (-13°C) is the main reason for the cold events (Figure 12l). Adiabatic warming is very similar between the two days, while diabatic heating (2.8°C) increases slightly, dampening the temperature decrease.

Similar to DJF, JJA cold events are characterized by a shift from continental to maritime air masses (Figs. S8c-d). The events are mainly triggered by cold air advection (-10.8°C), mitigated by increased diabatic heating (6.6°C), with adiabatic heating (1.3°C) changes being of minor importance (Fig. S8l).







**Figure 12. The spatial distribution of trajectories initiated on the previous day (t-1) and event day (t) for both December-February**
**(DJF) warm and cold events over Australia. In the top row, the color-shading illustrates the air parcel trajectory density (%)**
**based on the position between -5d and 0d. The magenta and yellow contours represent 0.5% particle density fields at -3d**
**and -1d, respectively. The red box shows the selected grid point over Australia. The Lagrangian evolution of distinct physical**



**parameters (pressure, temperature, potential temperature) along the air parcel trajectories for both warm (2nd row) and cold events (3rd row) is presented in panels e-j. Panels k and l show the contribution of the different physical processes to the genesis of**
**DTDT extremes according to Eq. (6), which refers to a 3d-time scale. The residuum is attributable to numerical inaccuracies in the computation of derivatives in Eq. (6). The box spans the 25th and 75th percentile of the data; the black dot inside the box gives the mean of the related quantities, and 1.5 times the interquartile range gives the whiskers.**

## 4. Discussion and summary

In this study, we have investigated (extreme) DTDT changes and the underlying physical processes. DTDT changes and
extremes have a larger magnitude in the extratropics compared to tropical regions during both DJF and JJA, consistent with previous studies (Xu et al., 2020; Zhou et al., 2020). These spatial patterns are associated mainly with differences in the standard deviation of daily temperature, but differences in the temporal autocorrelation also play a role for some regional variations. The patterns are generally comparable between ERA5 reanalysis data and HadGHCND observations, with typically larger magnitudes of DTDT changes in ERA5 due to both higher standard deviations and lower autocorrelation.
The mechanisms driving extreme DTDT changes (cold events below the 5th percentile and warm events above the 95th percentile) have been analyzed in detail for selected locations, using a combination of Eulerian composites, and Lagrangian process analysis to decompose the effects of advection, adiabatic and diabatic heating on a 3d-time scale. In the extratropics, extreme DTDT changes are typically associated with distinct synoptic circulation patterns, in particular troughs and ridges in the 500-hPa geopotential height field, similar to extremes of near-surface temperature (White et al., 2023; Parker et al., 2013;
Nygård et al., 2023). These patterns may be related to specific large-scale modes, such as the North Atlantic Oscillation, Artic Oscillation, or Southern Annular Mode, which significantly impact air mass advection (Lee et al., 2020; Liu et al., 2023; Dai and Deng, 2021). Such potential linkages can be studied in more detail in future research.

Our trajectory analysis shows that changes in advection are the main driver of extreme DTDT changes in the extratropics,
while the contributions of adiabatic and diabatic processes vary in space and also between warm and cold events. Changes in descent and adiabatic warming between the two days of an extreme DTDT change are either small or dampen the intensity of the events, for instance, for warm and cold events in Europe and South Asia (Figs. S10c-d) in JJA and in eastern and western North America and south Asia in DJF (Figs. S2a-b and S10a-b), with the exception of warm events in eastern North America, and high-latitude North Asia (Figs. S9c-d) in JJA and Europe in DJF, where they contribute positively. Diabatic
processes have a particularly strong effect on extreme DTDT changes in southern Australia, where they dampen the events' intensity in JJA but strongly amplify warm events in Austral summer. The latter is reminiscent of diabatic effects on wildfires and heatwaves in this region (Quinting and Reeder, 2017; Magaritz-Ronen and Raveh-Rubin, 2023). Apart from this, diabatic processes slightly amplify both warm and cold extremes in eastern North America, North Asia (Figs. S9a-b), and South Asia (Figs. S10a-b) during DJF, and primarily warm events in Europe and western North America (Figs. S6c-d)
during JJA. Comparing these processes associated with extreme DTDT changes with the mechanisms leading to usual temperature extremes (heat and cold waves) indicates similarities in the winter season when also temperature extremes are




strongly affected by advection in many mid-latitude regions (Bieli et al., 2015; Nygård et al., 2023; Röthlisberger and Papritz, 2023b; Kautz et al., 2022), but larger differences in summer, when extreme DTDT events are still primarily driven by advection, but especially heat waves in larger parts of the mid-latitudes are not (Zschenderlein et al., 2019; White et al., 2023).


We have studied the mechanisms associated with extreme DTDT changes in the tropics for a particular location at 13°S in South America and South Africa (Figs. S6 and S7) and found large differences between seasons. On the one hand, in JJA, advection is the main contributor to extreme DTDT changes, and interactions with the extratropics play a role, e.g., for the

inflow of colder air masses towards lower latitudes during cold events upstream of a subtropical trough. This configuration resembles the circulation impacting the cold waves in Central South America (Marengo et al., 2023) and South Africa (Chikoore et al., 2024) in JJA 2021. On the other hand, during DJF, extreme DTDT changes occur primarily due to local-scale diabatic processes associated with changes from cloudy conditions with precipitation to less cloudy and drier conditions or vice versa. This points to an important role of cloud radiative effects, primarily through the reflection of solar

radiation, for these events (cf. (Dai et al., 1999; Betts et al., 2013; Medvigy and Beaulieu, 2012).

Our study provides the first quantitative insights into the physical atmospheric processes that lead to extreme temperature changes from one day to another. The role of changing advection from warmer or colder regions for extreme DTDT changes is apparent across all studied regions except the tropics during DJF. Such advective effects are modified by Lagrangian

temperature changes due to adiabatic or diabatic processes, which tend to either amplify or dampen extremely positive (warm events) and negative (cold events) DTDT changes, depending on the region and season. This dominant effect of advection also explains why the magnitude of DTDT changes is typically larger in the extratropics, where horizontal temperature gradients and wind velocities are larger compared to the tropics. These mechanistic insights will be the basis for studying projected future changes in extreme DTDT changes in the second part of this study.

**Appendix**

The DTDT ($\delta_T$) change, as defined in Eq. (1), which has been determined based on the temperature 2m above the surface, is approximated through the average temperatures of the trajectories initiated on the corresponding day at their initiation time 0, denoted as $\delta_T^0$:

$$\delta_T \approx \delta_T^0 = \overline{T}_t^0 - \overline{T}_{t-1}^0$$

Note that the lower index here refers to the day on which the trajectories have been initiated, and the upper index refers to the time along the backward trajectory. These trajectory temperatures can then be expressed through the Lagrangian temperature evolution:

$$\delta_T^0 = \overline{T}_t^0 - \overline{T}_{t-1}^0$$





$$= \overline{T}_t^0 - \overline{T}_t^{-3d} - \overline{T}_{t-1}^0 + \overline{T}_{t-1}^{-3d} + \overline{T}_t^{-3d} - \overline{T}_{t-1}^{-3d}$$

$$= \Delta_{\overline{T},t} - \Delta_{\overline{T},t-1} + \delta_{\overline{T}}^{-3d} \tag{A1}$$

Here, $\delta_T^{-3d} = \overline{T}_t^{-3d} - \overline{T}_{t-1}^{-3d}$ measures the contribution due to advection and $\Delta_{\overline{T},t} = \overline{T}_t^0 - \overline{T}_t^{-3d}$ denotes the Lagrangian temperature change along the trajectories on a 3-day timescale.

The Lagrangian temperature change can be further decomposed into adiabatic and diabatic contributions using the thermodynamic energy Equation:

$$\frac{dT}{dt} = \frac{\kappa T \omega}{p} + \left(\frac{p}{p_0}\right)^{\kappa} \frac{d\theta}{dt} \tag{A2}$$

$\frac{dT}{dt}$ is the temperature change along the trajectory, $\kappa$=0.286, p denotes pressure ($p_0 = 1000\ hPa$), $\omega$ vertical velocity in pressure coordinates, and $\theta$ the potential temperature.

Accordingly:

$$\Delta_{\overline{T},t} = \langle \int_{-3d}^{0} \frac{dT_{t,i}}{d\tau} d\tau \rangle_i \tag{A3}$$

where $T_{t,i}$ is the temperature along the $i^{th}$ trajectory, $\langle \ldots \rangle_i = \sum_{i=1}^{m} \frac{1}{m} \ldots$ denotes the average over the m trajectories and $\tau$ is the time along the trajectory. Inserting (A3) into (A2) yields

$$\Delta_{\overline{T},t} = \Delta_{\overline{T},t}^{adi} + \Delta_{\overline{T},t}^{dia} \tag{A4}$$

with the adiabatic term $\left(\Delta_{\overline{T},t}^{adi}\right) = \langle \int_{-3d}^{0} \frac{\kappa T \omega}{p} d\tau \rangle_i$

$$= \langle \int_{p^{-3d}}^{p^0} \frac{\kappa T}{p} dp \rangle_i$$

and the diabatic term $\left(\Delta_{\overline{T},t}^{dia}\right) = \langle \int_{-3d}^{0} \left(\frac{p}{p_0}\right)^{\kappa} \frac{d\theta}{d\tau} d\tau \rangle_i$

$$= \langle \int_{\theta^{-3d}}^{\theta^0} \left(\frac{p}{p_0}\right)^{\kappa} d\theta \rangle_i$$

Then Eq. (A1) becomes

$$\delta_T^0 = \delta_{\overline{T}}^{-3d} + \Delta_{\overline{T},t}^{adi} - \Delta_{\overline{T},t-1}^{adi} + \Delta_{\overline{T},t}^{dia} - \Delta_{\overline{T},t-1}^{dia}$$

$$= \delta_{\overline{T}}^{-3d} + \delta_{\overline{T}}^{adi} + \delta_{\overline{T}}^{dia} \tag{A5}$$

In Eq. (A5), $\delta_T^0$ is the approximated near-surface DTDT change, $\delta_{\overline{T}}^{-3d}$ measures the mean temperature difference at the origin of the air parcels and thus the contribution of advection on a 3-day time scale, $\delta_{\overline{T}}^{adi}$ is the mean temperature difference created through adiabatic compression or expansion resulting from vertical descent or ascent, respectively, and $\delta_{\overline{T}}^{dia}$ is the contribution of mean diabatic heating or cooling from processes such as latent heating in clouds, radiation, and surface fluxes.



**Code and data availability**

The code of the trajectory model LAGRANTO is available at https://iacweb.ethz.ch/staff/sprenger/lagranto/download.html (Sprenger and Wernli, 2015). The HadGHCND data used in this study can be freely accessed from http://www.metoffice.gov.uk/hadobs/hadghcnd/download.html. ERA5 data are available via the Copernicus Climate Change

Service (C3S; https://doi.org/10.24381/cds.143582cf; Hersbach et al., 2018).

**Author contributions**

Both authors designed the study. KH performed the analysis, produced the figures, and drafted the manuscript. Both authors discussed the results and edited the manuscript.

**Competing interests**

Stephan Pfahl is the executive editor of WCD.

**Acknowledgments**

We acknowledge the HPC service of ZEDAT, Freie Universität Berlin, for providing computational Resources (Bennett et al., 2020).

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
