# Peer review of "Physical Processes Leading to Extreme Day-to-Day Temperature Change, Part I: Present-day Climate"

_EGUsphere, 2024_

## Referee Comment (RC1)

**Review of**
**"Physical Processes Leading to Extreme day-to-day Temperatures Changes, Part 1: Present-day Climate"**
**by Kalpana Hamal and Stephan Pfahl**
**submitted to Weather and Climate Dynamics**

**General comment:**

This manuscript investigates the mechanisms behind extreme day-to-day warming and cooling events at four locations around the globe. To this end, changes in the mean circulation patterns at these locations are examined and variations in the behavior of air parcel trajectories are identified. That way, the study quantifies the contributions of transport (advection), adiabatic heating, and diabatic heating to extreme day-to-day temperature changes. The authors find that advection is the main factor driving extreme temperature changes in the extratropics, whereas advection is of lesser importance for extreme temperature changes in the tropics. I find the study interesting, especially because it focuses on extreme temperature changes rather than extreme temperatures themselves, offering a new perspective on temperature variability in the atmosphere. I do not have significant scientific concerns. However, I see considerable potential for improving the language used. I feel that often inappropriate or too imprecise phrasing detracts from the content and makes it difficult to fully grasp the intended message. I recommend a thorough revision of the manuscript, with each sentence being carefully reviewed for clarity and logical flow. Therefore, I suggest publication after major revisions. Below, I have compiled a list of questions and some ideas for improving the text.

**Major comments:**

1) To me, it is often unclear whether the text is addressing temperature extremes or extreme temperature variability. I believe this is due to imprecise terminology. For example, throughout the text, the terms "warm events" and "cold events" are used to describe extreme day-to-day temperature change events. However, I find these expressions somewhat misleading, as they may imply that you are looking at warm and cold temperature extremes themselves, rather than on extreme warming and cooling events. The text would be much clearer if "warm events" were replaced with "warming events" and "cold events" with "cooling events." The same is true for the expression "DTDT extremes", which I think should be better replaced by something like "extreme DTDT changes". Furthermore, at some points (introduction, summary), you make a connection between your work and studies focusing on the mechanisms behind the development of warm and cold temperature extremes. I think it should be made clearer at this point that an extreme temperature event does not necessarily have to be linked to extreme temperature variability. A sudden temperature increase does not necessarily occur when it gets particularly hot, and similarly, a sharp temperature drop does not have to happen when it is particularly cold.

2) Overall, I feel that the term "advection" is not applied with sufficient precision. At times, it is used when showing instantaneous wind fields (e.g. L210, L2022), and at other times when discussing trajectories and the transport of air masses (e.g. L211, L217). I believe it is crucial to be very careful about when the term "advection" is used and when it might be more appropriate to use a different term, especially considering that the existing literature is not always clear on this matter. Furthermore, I think it would be very helpful to clarify what is meant by "cold air advection" and "warm air advection," i.e. with respect to what is the air cold or warm. I believe what you mean is that, for instance, the air transported to the location on the day of a cooling event is originally colder than the air that was transported there the previous day. I think it would be very beneficial to be more precise here.

3) My last major comment relates to the adiabatic warming. Whenever you describe the temperature at t-1, you write something similar to L216: "Accordingly, the temperature at t-1 is mainly determined by cold air advection, mitigated by adiabatic warming, ...". I was wondering if it is not always the case that an air mass at the surface undergoes adiabatic warming. If so, is it truly necessary—or perhaps redundant—to specify each time that the temperature has been affected by some adiabatic warming? The same applies to the initial temperature, which is always lower than the final temperature. I believe the text could be condensed on this point such that greater emphasis on the differences between day t-1 and day t is given.

**Minor comments:**

L102-117: This paragraph is very hard to follow. I think this is because the expressions used in the formulas are not well described and because equations (3) and (4) are never explained in the text. I suggest to break down the equations to what is really important and to introduce the equations step by step, instead of a full block of equations.

L121-122: The number of identified events should differ between the ERA5 data and the HadGHCND data, since their used periods differ. To which dataset do the numbers refer to?

L131: To better understand the underlying mechanism of what?

L133: Instead of "apply a novel" "introduce a novel" to make clear that this precise decomposition has not been used before?

L133: Lagrangian temperature **variability** decomposition?

L137: The phrase "The Lagrangian decomposition of DTDT changes, as approximated by the trajectories" is odd, since the trajectories do not approximate the Lagrangian decomposition. Rather, the Lagrangian decomposition is obtained from computed trajectories.

L 141: What is decomposed?

L142: I think at this point it would be very worth noting that advection in this approach refers to something different than in the approach by Röthlisberger and Papritz (relates to major comment 2)

L149: instead of "magnitude of σ changes" either "magnitude of DTDT variations, quantified by σ", or simply " magnitude of σ"

L150-153: Here it is written "the variability is larger during DJF than in JJA", followed by "the variability is above 3 °C in DJF compared to 1-4 °C in JJA". However, 4 °C is larger than 3 °C, such that one could conclude that the variability in JJA is larger than in DJF. Please rephrase this sentence more precisely.

L150-153: Here it is shown that the magnitude in DTDT changes is larger in DJF than in JJA, irrespective of the hemisphere. This means that in the northern hemisphere, the magnitude of DTDT changes is larger in winter than in summer. In contrast, in the southern hemisphere, the magnitude of DTDT changes is larger in summer than in winter. Is this behavior expected? Can you think of any explanations for this behavior?

L153: rephrase "remain consistently"

L153/L155 and other lines: Similar to comment to L149: To my understanding the phrases "σ variations" and "σ changes" do not make sense. I think it would be more accurate to simply use "σ".

L159-161: "Since the magnitude of σ changes can be expressed as a function of …, Figures 1 and 2 show these related quantities." Again, I think this sentence is not properly formulated. The Figures 1 and 2 do not show the other quantities because σ can be expressed as a function of them. It is rather that you decided to show them as they are part of the computation of σ.

Figure 1/Figure 2: It would be helpful to use the same colorbars in Figure 1 and Figure 2 to enable an easier comparison.

L183/185: I think it is incorrect to use the phrase "leads to" here. Replace by something like "associated with".

L184: "smaller" instead of "lower"?

L209: I feel that the phrase "southerly airflow around its western flank" is somewhat misleading as it suggests that you refer to trajectories/air parcels. But what is shown in Figure 4a is the wind.

L211: Similar to the previous comment. I do not think that you can really see "cold air advection" in this plot. You see northerly winds blowing across a temperature gradient, suggesting cold air advection.

L207/L214: I was wondering whether the word "distinct" is appropriate here.

L214: What is meant by "limited" diabatic cooling?

L215-216: I suggest mentioning once that the residual is small, e.g., as the last sentence in the section "Lagrangian Temperature Decomposition," and then omitting it in the following text and the figures.

L222: Again, I think the use of the word advection is somewhat misleading here. I suggest to use "southwesterly wind" instead of "southwesterly advection" (se major comment 2).

Caption Figure 4: You write "selected grid point" but shown is a "grid box".

L249-267: To shorten the entire paragraph: Could you simply say that a DJF cooling event is essentially the same as a "reversed" DJF warming event? To me, Figures 4a and 4d look quite similar, as well as Figures 4b and 4c. And Figure 5k is more or less the same as Figure 5l, just mirrored.

L285: I suggest to cross the "which are only briefly discussed", since it gets clear from the "focuses on JJA" that the focus is not on DJF.

L333: For instance, at this point it would be very beneficial to be precise of what is meant by cold air advection and warm air advection, i.e. with respect to what is the air cold or warm (see major comment 2).

L356/L385/L385/L493: What do you mean with "distinct" patterns and "specific" circulation patterns? Do you mean that all events exhibit a similar pattern, e.g., in the 500 hPa geopotential height? If so, I think you cannot deduce that from the plots, as you only show the mean circulation across all events, which might differ substantially from one event to another.

L356-373: Are you referring to cooling or warming events here?

L387-388: At this point, I think one must be very careful with the phrasing. You are looking at a budget, so you cannot say that term 1 and term 2 cancel each other out, leaving only term 3 as important! Imagine term 1 being +5 K, term 2 being -5 K, and term 3 being +5 K. You cannot say that term 1 and term 2 cancel each other out, leaving only term 3 as important, nor can you say that term 2 and term 3 cancel each other out, leaving only term 1 as important.

L416-425: Again, to shorten the paragraph: Could you simply say that DJF cooling events are essentially the same as a "reversed" DJF warming event?

L393: I suggest to write "… **presumably** contributing to larger diabatic heating and higher temperatures"

L500: To better connect to the first part of the sentence, you could insert something like "while the contributions of adiabatic and diabatic processes **are generally smaller and** vary **more** in space and also between warm and cold events".

L513/514: Here, scientific debate is still going on. There are also studies saying that heat waves in the mid-latitudes are driven by advection (e.g., Harpaz et al. 2014, Sousa et al. 2019, partly Röthlisberger and Papritz 2023).

L537: I do not understand what is meant by the sentence "is approximated through the average temperatures of the trajectories initiated on the corresponding day at their initiation time".

**Technical corrections:**

Title: temperature without plural?

L120: 5th and 95th percentiles of the DTDT change **distribution**

L125,136: no plural: DTDT change event?

L190: DTDT **variations**

L193: DTDT **variation** events

L194: DTDT variations

Figures: I think it would be helpful to keep the direction of panel labeling consistent across all figures.

L202/284/353/…: invest the mechanim**s**?

L220: On the day**s** of the ...

L241: smaller **in** magnitude?

L339: compared **to**

L445: result from

---

## Referee Comment (RC2)

**Review of "Physical Processes Leading to Extreme day-to-day Temperatures Changes, Part I: Present-day Climate"**

The work of Hamal and Pfhal investigates the physical processes of extremes in the day-to-day variability of near-surface temperature (DTDT) using the ERA5 data set. They use both composites maps and Lagrangian backward trajectories tracking to quantify the mechanisms for these extreme changes in temperature. Their main conclusion is that these extremes are mainly driven by changes in advection in the extra-tropics while the situation is more balanced in the tropics with important contributions from diabatic processes.

The paper is well written and the research question is clearly stated and interesting. The analyses carried out in the paper are careful, well explained and scientifically sound. I should say that the results are not particularly jaw-dropping – I would have guessed that advection was the main contributor of extremes of DTDT in the extra-tropics – but they are nonetheless interesting for documenting these mechanisms. I have some suggestions to improve the quality of the communications of the results and some additional analyses. Therefore I recommend major revisions at this stage, see below for my comments.

**Major comments**

**1. Use of ERA5 and HadGHCND:** I think the comparison between the DTDT variability in ERA5 and HadGHCND is problematic. As the authors show in their Figure 1, there are large differences between the two data sets that are probably not physical. The reason is likely because HadGHCND interpolates station data to construct a gridded data set which likely smooths out the daily variability (rather than lack of station coverage I think). In my opinion, this makes the HadGHCND data set particularly not suited for the study that you are doing here. That being said, it is true that the authors mainly compare the spatial patterns rather than the absolute values of sigma_DTDT between the two data sets. If you really want to compare the absolute values found with ERA5 with measured data you should probably go directly to station data. For these reasons, I would discourage to show the comparison with HadGHCND in the figures of the main text: the authors can include it in supplementary materials if they really want to do this comparison. In this case they should also discuss more the differences between the two data sets. Moreover, the rest of the paper does not use HadGHCND.

**2. Statistical suggestions:**
      a. I think it would be interesting to show (at least for the grid points studied) the distribution of delta_T and the quantiles that you are selecting. In particular it would be interesting to see whether the distribution is symmetric. You could for example compute, in addition to its standard deviation, its kurtosis and show the corresponding map.
      b. One question I had while reading the paper is how much the extremes of delta_T relate to extremes of T, in other words: do your warming/cooling events also correspond to warm/cool extremes? I think it would be super interesting to show how the extremes of delta_T are linked to the quantiles of T_t and T_t-1. For example, do extreme warming events happen because we start from a very cold quantile and we end up in the middle of the temperature distribution or do we start from the middle of the distribution and end up in the right tail ? The physical processes in these two situations are likely different.

3. **Comparison with climatology:** I find it really interesting that in Figure 4-5 and others the warming and cooling events seem the reverse of one another. As advection seems to be the largest contributor, it seems to me that extremes of sigma_DTDT happens as if this mechanism was switched on or off: warming events happen because the northward advection was switched off and vice versa for cold events. This leads me to my question

which is not unrelated to my previous comment 2.a., how are the dynamical situations that you identify unusual with respect to climatology ? Is it the starting point that is dynamically unusual or the end point ? To be more clear, it seems to me that you should probably do your composite maps also in anomalies.

**4. Extremes of DTDT and fronts:** the fact that advection is the main factor of extremes in DTDT in the extra-tropics is not super surprising and that is what I would expect because of the existence of atmospheric fronts (some may even argue that fronts are by definition extremes of DTDT). I am surprised that the authors do not mention at all these structures. Can you say a word about how your analysis and results relate to the literature on frontal structures ? Moreover, it seems to me that frontal structures are well identified in the climate variability literature as being the mechanism for day-to-day variability (e.g. Ghil and Lucarini (2020)), maybe you could also mention that in the broader context of your work.

**Minor comments**

1. Paragraph L49: in this paragraph you are mainly talking about hot and cold extremes, which are rather different from the extremes in DTDT that you are looking at. This may be confusing for the reader, please be more clear about how the extremes per se relate to the extremes of DTDT (see also my major comment 2).

2. For clarity, I think you should detail a bit more the terms in eq 1-5. In particular, equation 4 is not necessary to me and may be confusing. Moreover, you should explain what the approximation means in equation 3 (explain why this is actually a very good approximation and the errors involved are small because of the typical time scale of the seasonal cycle).

3. L126: can you detail a bit more why those choices were made, especially the date and time of the initialization of the backward trajectory.

4. Equation 6: this is more for my understanding: given that you are looking at air parcels close to the ground, how can the adiabatic contribution be anything else than positive ?

5. Figure 1: the colors scale in all panels is unfortunate. You are showing only positive, non-divergent values therefore you should not use a divergent color maps which is misleading for the reader. Also, because you compare between panels a,b and c,d, the values of the color map should have the same range. Finally, you should probably use the Robinson projection.

6. L149 and following: you are mentioning the "magnitude of sigma_DTDT changes". I am not sure what this is referring to, if I understood correctly you should rather talk about the "magnitude of sigma_DTDT".

7. L174: what are "the deep tropics" ?

8. Because you are studying land grid points only and their proportion varies a lot between the latitudes, I am not sure the zonal means in Figure 1 and following are really relevant: the reader can see by themselves that there is a marked latitudinal gradient of the quantities you are displaying. Also, do you have an explanation for why in the southern hemisphere the variability is much smaller than in the northern hemisphere for the grid points with the same absolute latitude ?

9. L205: "DJF warm events": I would strongly discourage you to use this phrasing for the events you are studying because it is really misleading for the reader. You should rather talk about warming/cooling events.

10. Fig4:
    a. Please define more precisely near surface temperature: is it T2M ?
    b. I would suggest not to use absolute values for composite maps: first because there is still the seasonal cycle (how do you handle that by the way ?) and second because it is difficult to read and the sudden change of temperature from t-1 to t is not very clear. Maybe simply use anomaly maps and/or make a difference between the map at t and the map at t-1 ?

11. Figure 5 and alike: it would be clearer for the reader if you could indicate explicitly on the figure (not only in the legend) if those are DJF or JJA events.

12. L431: "To systematically investigate the mechanism driving DTDT extremes over the subtropics in the southern hemisphere during DJF and JJA, we select a specific location in Australia": to me this sentence sounds self-contradictory, how can you systematically investigate if you look at only one grid point ?

13. The conclusions reached are based on the analysis of only some grid points at various longitudes/latitudes. Although I think the conclusions reached can probably be extended to the other grid points in the vicinity, I think the authors should be a bit more cautious in their concluding statements.

14. L531: "This dominant effect of advection also explains why the magnitude of DTDT changes is typically larger in the extratropics, where horizontal temperature gradients and wind velocities are larger compared to the tropics": this statement is likely true but deserves more evidence.

**References**

Ghil, M., & Lucarini, V. (2020). The physics of climate variability and climate change. *Reviews of Modern Physics*, *92*(3), 035002.

---

## Community Comment (CC2)

**I am very happy to read your work! Interestingly, I have conducted similar research using Lagrangian analysis, focusing on a case study of weather whiplash (https://doi.org/10.1088/1748-9326/ad9c9a). Back to your work, I have several questions and one possible suggestion that I would like to discuss with you!**

Thank you so much for your insightful questions and suggestions regarding our paper. We've read your paper on the weather whiplash case study and found it to be very exciting. It's fascinating to see how Lagrangian analysis can be applied to the whiplash case study as well. Below, we've provided answers to the questions you raised in our paper.

1. **Question-1: Your work investigates the extreme day-to-day temperature (DTDT) variations and offers a physical understanding via a Lagrangian temperature budget. While Lagrangian analysis vividly depict synoptic motions, why not employ the "Eulerian temperature budget" instead? This approach directly examines the contributions of different physical processes to DTDT variations, i.e., partial T/partial t.**

   **Answer:** The Eulerian temperature budget focuses on changes in temperature at fixed spatial locations, breaking these changes down into contributions from various physical processes. While this approach is valuable for assessing local processes and may be well-suited for tropical regions during December–January–February, it may not capture the movement, evolution, and associated temperature changes of air parcels, particularly in the extratropics. As the advection effects are modulated by adiabatic and diabatic processes within transported air parcels, amplifying or dampening DTDT decreases (cold events) and increases (warm events). For instance, during warm events in North America, remote advection plays a significant role, with some additional contributions from surface heating. In contrast, in Australia, diabatic heating during transport is more dominant than local heating. In an Eulerian budget, this diabatic heating of the air parcels would not have been accounted for, since only local heating is captured. This underscores the importance of the Lagrangian framework for studying extreme DTDT by tracking air parcels and their trajectories, offering a comprehensive understanding of the underlying dynamics.

2. **Question-2: Your manipulation leading to Eq. A1 is very interesting. However, the "advection term" in Eq. A1 does not seem to have an intuitively clear connection to advection. Could you clarify this?**

**Answer:** $\overline{T}_t^{-3d}$ indicates the average temperature of the air parcels initialized on the day of the extreme events three days before their arrival at the target location and $\overline{T}_{t-1}^{-3d}$ the corresponding temperature for the air parcels initialized one day earlier. The expression $\delta_T^{-3d} = \overline{T}_t^{-3d} - \overline{T}_{t-1}^{-3d}$ thus represents the difference between the air parcel's temperatures three days before their arrival. Assuming that no further temperature changes occurred during the transport, the DTDT change would only be due to these initial differences of the air parcels, which means it would be caused by changes in the advection of air parcels with different original temperatures between the previous day and the day of the event. This is why we refer to this term as an advection term.

**Suggestion-1: A study by Prof. Tapio Schneider provides a macroturbulence perspective (temperature gradient + mixing length) to understand synoptic temperature variability and its responses to global warming (https://doi.org/10.1175/JCLI-D-14-00632.1). It might be valuable to compare your findings with this work.**

**Answer:** Thank you for the suggestion. This paper investigates the underlying physical processes of DTDT extremes and their linkage to the large-scale atmospheric circulation in the present climate. We will also explore how these processes influence projected DTDT extremes under climate warming in Part II. The study by Prof. Tapio Schneider shows that the reduction of meridional potential temperature gradients, driven by the polar amplification of global warming, leads to a decrease in synoptic temperature variance near the surface. Comparing our findings with this work will offer additional insights into the dynamics of temperature extremes and their response to global warming.

---

## Author Comment (AC1)

Review of "Physical Processes Leading to Extremes Day-to-day Temperatures Change, Part 1: Present-day Climate"

by Kalpana Hamal and Stephan Pfahl submitted to Weather and Climate Dynamics

**General comments**

This manuscript investigates the mechanisms behind extreme day-to-day warming and cooling events at four locations around the globe. To this end, changes in the mean circulation patterns at these locations are examined and variations in the behavior of air parcel trajectories are identified. That way, the study quantifies the contributions of transport (advection), adiabatic heating, and diabatic heating to extreme day-to-day temperature changes. The authors find that advection is the main factor driving extreme temperature changes in the extratropics, whereas advection is of lesser importance for extreme temperature changes in the tropics. I find the study interesting, especially because it focuses on extreme temperature changes rather than extreme temperatures themselves, offering a new perspective on temperature variability in the atmosphere. I do not have significant scientific concerns. However, I see considerable potential for improving the language used. I feel that often inappropriate or too imprecise phrasing detracts from the content and makes it difficult to fully grasp the intended message. I recommend a thorough revision of the manuscript, with each sentence being carefully reviewed for clarity and logical flow. Therefore, I suggest publication after major revisions. Below, I have compiled a list of questions and some ideas for improving the text.

We would like to thank the reviewer for their helpful comments. Our responses are printed in blue whereas the reviewer questions are in black. In addition to addressing the individual comments, we will review the manuscript for clarity and flow.

**Major comments**

1. To me, it is often unclear whether the text is addressing temperature extremes or extreme temperature variability. I believe this is due to imprecise terminology. For example, throughout the text, the terms "warm events" and "cold events" are used to describe extreme day-to-day temperature change events. However, I find these expressions somewhat misleading, as they may imply that you are looking at warm and cold temperature extremes themselves, rather than on extreme warming and cooling events. The text would be much clearer if "warm events" were replaced with "warming events" and "cold events" with "cooling events." The same is true for the expression "DTDT extremes", which I think should be better replaced by something like "extreme DTDT changes". Furthermore, at some points (introduction, summary), you make a connection between your work and studies focusing on the mechanisms behind the development of warm and cold temperature extremes. I think it should be made clearer at this point that an extreme temperature event does not necessarily have to be linked to extreme temperature variability. A sudden temperature increase does not necessarily occur when it gets particularly hot, and similarly, a sharp temperature drop does not have to happen when it is particularly cold.

   **Response:** Thank you for your feedback. Throughout the paper, we will use 'warming events' and 'cooling events' instead of 'warm events' and 'cold events,' which improves clarity. We will also rephrase 'DTDT extremes' as 'extreme DTDT changes' to avoid ambiguity.

   We appreciate your comment and agree that it is important to clarify the distinction between extreme temperature events and extreme DTDT changes. While we acknowledge that extreme temperature events and extreme DTDT changes are not necessarily linked, our focus on daily temperature extremes in the introduction provides context for atmospheric circulation patterns, particularly given the limited literature on extreme DTDT variability.

Our analysis, as also indicated by Equation (5), shows that variations in $\sigma_T$ primarily determine the spatial variability of $\sigma_{DTDT}$. Additionally, we explore the relationship between DTDT changes and specific quantiles (deciles) of $T_t$ and $T_{t-1}$, as illustrated in Figure R1. Extreme warming events are associated with changes from lower-middle temperature quantiles ($T_{t-1}$) to middle-higher temperature quantiles ($T_t$), while the reverse is true for cooling events. Thus, extreme DTDT changes typically occur when $T_t$ predominantly falls within the tail quantiles (stippling in Figure R1).

$$\sigma_{DTDT} = \sigma_T \sqrt{2\,(1\text{-}r_{1,T})} \qquad\qquad (5)$$

[Figure]

**Figure R1. Heatmaps of the relationship between DTDT change and the deciles of temperature on the previous day ($T_{t-1}$) and the event day ($T_t$) for December-February (DJF) and (b) June-August (JJA) for North America. The x-axis and y-axis represent deciles of $T_t$ and $T_{t-1}$, while the color shading indicates DTDT changes, with red and blue colors indicating warming and cooling, respectively. The black circles represent extreme DTDT changes.**

To further illustrate the differences in atmospheric circulation between extreme DTDT changes and daily extreme events, for example, the corresponding composites for the winter season and the selected location in North America are shown in Figure R2. They reveal similar circulation patterns, which, however, are more pronounced in the case of daily temperature extremes (Figure R2). Similar results are observed at other grid points. We will add this analysis as a supplementary.

[Figure]

**Figure R2. Composite of near-surface temperature (T2M, °C, color shading), wind at 850 hPa (UV, m/s, vectors), and Geopotential height at 500 hPa (GP, gpm, magenta contours) on the (a, c) previous day (t-1) and (b, d) event day (t ) of warming (a,b) and cooling events (c,d) during December-February (DJF) at a selected grid in North America (red box). Composites of (e) warm and (f) cold events, defined with the 5th and 95th percentiles of the daily temperature distribution as thresholds, are shown in the third row, and the difference between daily extremes (panels e and f) and DTDT extremes (panels c and d) are shown in panels g-h. Note that in a-f wind vectors ≥5m/s and in g-h wind vector anomalies ≥1m/s are plotted. The dotted and bold magenta contours in g-h indicate negative and positive values of geopotential, respectively.**

2. Overall, I feel that the term "advection" is not applied with sufficient precision. At times, it is used when showing instantaneous wind fields (e.g. L210, L2022), and at other times when discussing trajectories and the transport of air masses (e.g. L211, L217). I believe it is crucial to be very careful about when the term "advection" is used and when it might be more appropriate to use a different term, especially considering that the existing literature is not always clear on this matter. Furthermore, I think it would be very helpful to clarify what is meant by "cold air advection" and "warm air advection," i.e. with respect to what is the air cold or warm. I believe what you mean is that, for instance, the air transported to the location on the day of a cooling event is originally colder than the air that was transported there the previous day. I think it would be very beneficial to be more precise here.

**Response:** Thank you for the suggestion. We will update the usage of the term advection and clarify the cold advection and warm advection terms in the methodology. To avoid confusion, the term will be used exclusively in the discussion of air mass transport (trajectories) and not anymore for wind composites.

Technically, the advection term in our Lagrangian budget is defined as follows: As most temperature extremes were found to develop within a 2–3-day timescale (Bieli et al., 2015), we selected 3-day backward trajectories for the budget calculations in our study. $\overline{T}_t^{-3d}$ represents the average temperature of the air parcels initialized on the day of the extreme event, three days before their arrival at the target location, while $\overline{T}_{t-1}^{-3d}$ represents the corresponding temperature for the air parcels initialized one day earlier. The expression $\delta_T^{-3d} = \overline{T}_t^{-3d} - \overline{T}_{t-1}^{-3d}$ thus captures the difference in temperature between the air parcels three days before their arrival. Assuming no further temperature changes occurred during transport, the DTDT change is solely due to these initial differences. This suggests that variations in the advection of air parcels with different original temperatures between the previous day and the day of the event cause the temperature changes. This is why we refer to this term as an advection term. When the temperature of the air parcels initialized on the previous day ($\overline{T}_{t-1}^{-3d}$) is higher than the temperature of the air parcels at the event ($\overline{T}_t^{-3d}$), this represents a shift from warmer air to colder air on the event day, which is referred to as cold air advection. The reverse is true for warm air advection.

3. My last major comment relates to adiabatic warming. Whenever you describe the temperature at t-1, you write something similar to L216: "Accordingly, the temperature at t-1 is mainly determined by cold air advection, mitigated by adiabatic warming, ...". I was wondering if it is not always the case that an air mass at the surface undergoes adiabatic warming. If so, is it truly necessary—or perhaps redundant—to specify each time that the temperature has been affected by some adiabatic warming? The same applies to the initial temperature, which is always lower than the final temperature. I believe the text could be condensed on this point such that greater emphasis on the differences between day t-1 and day t is given.

**Response:** Thank you for the suggestion. Our approach involves studying the temperature evolution on each of the two days individually, followed by analyzing their differences. While the air masses always undergo adiabatic warming (since they arrive near the surface), the magnitude of this warming can be different, with some air masses descending more than others. Accordingly, the contribution to DTDT changes can be both negative or positive, depending on typical differences in the strength of the descent between the two days. Our results indicate that the mean effect of such changes in adiabatic warming is relatively small in many regions, but they contribute substantially to event-to-event variability.

Nevertheless, we understand the reviewer's point that the text should mainly focus on explaining the differences between the two days that make up the DTDT changes, and we will thus further condense the descriptions of the evolution during the individual days.

**Minor comments**

1. L102-117: This paragraph is very hard to follow. I think this is because the expressions used in the formulas are not well described and because equations (3) and (4) are never explained in the text. I suggest to break down the equations to what is really important and introducing the equations step by step, instead of a full block of equations.

   **Response:** We will explain equations 1-5 in more detail:

   This study defines DTDT change, denoted as $\delta_T$, as the difference in daily mean near-surface air temperature between the previous day ($T_{t-1}$) and the day of the event ($T_t$), as shown in Eq. (1).

   $$\delta_T = (T_t - T_{t-1})  \qquad (1)$$

   The average daily temperature change, $\mu_{DTDT}$ reflects the difference between the temperatures at the start ($T_0$) and end ($T_n$) of the time series (Eq. 2).

   $$\mu_{DTDT} = \frac{1}{n}\sum_{t=1}^{n}(T_t - T_{t-1}) = T_n - T_0  \qquad (2)$$

   To capture typical day-to-day temperature changes, we thus use the standard deviation, $\sigma_{DTDT}$, as shown in Eq. (3).

   $$\sigma_{DTDT}^2 = \frac{1}{n}\sum_{t=1}^{n}(T_t - T_{t-1})^2  \qquad (3)$$

   By inserting the average daily temperature $\mu_T$ and multiplying out the square bracket, we find a relationship between $\sigma_{DTDT}$, the standard deviation of the daily mean temperature ($\sigma_T$) and the covariance between consecutive days (COV ($T_t, T_{t-1}$)):

   $$\sigma_{DTDT}^2 = \frac{1}{n}\sum_{t=1}^{n}((T_t - \mu_T) - (T_{t-1} - \mu_T))^2$$

   $$= \frac{1}{n}\sum_{t=1}^{n}((T_t - \mu_T)^2 + (T_{t-1} - \mu_T)^2 - 2(T_t - \mu_T)(T_{t-1} - \mu_T)$$

   $$\approx 2\sigma_T^2 - 2\,COV\,(T_t, T_{t-1})  \qquad (4)$$

   The approximation in equation (4) is associated with the fact that, for large n, both $\frac{1}{n}\sum_{t=1}^{n}(T_{t-1} - \mu_T))^2$ and $\frac{1}{n}\sum_{t=1}^{n}(T_t - \mu_T))^2$ are good estimators of $\sigma_T^2$. Finally, the standard deviation of DTDT can thus be expressed as a function of the usual standard deviation ($\sigma_T$) and the lag-1 autocorrelation $r_{1,T}$ of daily mean temperature, as shown in Eq. (5).

   $$\sigma_{DTDT} = \sigma_T\sqrt{2\,(1 - r_{1,T})}  \qquad (5)$$

2. L121-122: The number of identified events should differ between the ERA5 data and the HadGHCND data, since their used periods differ. To which dataset do the numbers refer to?

   **Response:** Yes, it is true that the number of events differs depending on the choice of datasets. We used HadGHCND data for the comparison of DTDT variability patterns with ERA5 in Figures 1 and 2. For extreme DTDT events, we only used the ERA5 dataset, as shown in Figure 2 (in the paper). The sentence will be updated accordingly in the paper.

3. L131: To better understand the underlying mechanism of what?

   **Response:** We will add the following: "To better understand the underlying mechanisms of extreme DTDT changes"

4. L133: Instead of "apply a novel" "introduce a novel" to make clear that this precise decomposition has not been used before?

   **Response:** We will change this.

5. L133: Lagrangian temperature variability decomposition?

   **Response:** We will change this.

6. L137: The phrase "The Lagrangian decomposition of DTDT changes, as approximated by the trajectories" is odd, since the trajectories do not approximate the Lagrangian decomposition. Rather, the Lagrangian decomposition is obtained from computed trajectories.

   **Response:** We will change this.

7. L 141: What is decomposed?

   **Response:** The sentence will be improved as "The DTDT change has been decomposed into three contributing factors"

8. L142: I think at this point it would be very worth noting that advection in this approach refers to something different than in the approach by Röthlisberger and Papritz (relates to major comment 2).

   **Response:** Yes, we have used a different method to calculate the advection term than Röthlisberger and Papritz (2023). For daily temperature extremes, they use horizontal advection of the air parcel in the direction of the climatological temperature gradient. As we did not decompose the temperature into anomalies and climatology (which is less useful for the analysis of DTDT changes, for which the temperature is not necessarily very anomalous on both involved days), we have used a simpler definition, as explained in our response to the second major comment. This explanation will be added to the revised manuscript.

9. L149: instead of "magnitude of σ changes," either "magnitude of DTDT variations, quantified by σ", or simply "magnitude of σ"

   **Response:** We will improve the sentence as "Both the HadGHCND and ERA5 datasets reveal that the magnitude of DTDT variations, quantified by $\sigma_{DTDT}$ is larger…"

10. L150-153: Here it is written "the variability is larger during DJF than in JJA", followed by "the variability is above 3 °C in DJF compared to 1-4 °C in JJA". However, 4 °C is larger than 3 °C, such that one could conclude that the variability in JJA is larger than in DJF. Please rephrase this sentence more precisely.

    **Response:** We will rephrase to "In DJF, $\sigma_{DTDT}$ is above 4°C in many regions (Figures 1a-b), compared to 1-4°C during JJA (Figures 2a-b)".

11. L150-153: Here it is shown that the magnitude in DTDT changes is larger in DJF than in JJA, irrespective of the hemisphere. This means that in the northern hemisphere, the magnitude of DTDT changes is larger in winter than in summer. In contrast, in the southern hemisphere, the magnitude of DTDT changes is larger in summer than in winter. Is this behavior expected? Can

you think of any explanations for this behavior?

**Response:** We expect this behavior during the northern hemisphere winter, as the atmospheric circulation is stronger (Figure 4 in the paper). As anticipated, both warming and cooling events in the hemisphere are primarily driven by advection. In the southern hemisphere, the seasonality is less clear and depends on the region and dataset. We will mitigate our statement in the manuscript accordingly. For our selected location in southern Australia, the large DTDT changes during the DJF season (summer for this region), are due to a more dominant role of diabatic processes associated with stronger heating of the air masses over the continent due to stronger surface fluxes in summer.

12. L153: rephrase "remain consistently"

**Response**: We will rephrase to "However, $\sigma_{DTDT}$ is around 1-2°C in the tropics and 1-3°C over higher-latitude land regions in the southern hemisphere…."

13. L153/L155 and other lines: Similar to comment to L149: To my understanding the phrases "$\sigma$ variations" and "$\sigma$ changes" do not make sense. I think it would be more accurate to simply use "$\sigma$".

**Response:** We will change this.

14. L159-161: "Since the magnitude of $\sigma$ changes can be expressed as a function of …, Figures 1 and 2 show these related quantities." Again, I think this sentence is not properly formulated. The Figures 1 and 2 do not show the other quantities because $\sigma$ can be expressed as a function of them. It is rather that you decided to show them as they are part of the computation of $\sigma$.

**Response:** The sentence will be reformulated: "According to equation 5, the magnitude of DTDT temperature changes can be expressed as a function of the standard deviation $\sigma_T$, and lag-1 autocorrelation $r_{1,T}$ of daily mean temperature, which are also shown in Fig. 1 and 2."

15. Figure 1/Figure 2: It would be helpful to use the same colorbars in Figure 1 and Figure 2 to enable an easier comparison.

**Response:** We will adapt the color bar for easier comparison in the revised manuscript.

16. L183/185: I think it is incorrect to use the phrase "leads to" here. Replace by something like "associated with".

**Response:** We will change this.

17. L184: "smaller" instead of "lower"?

**Response:** We will change this.

18. L209: I feel that the phrase "southerly airflow around its western flank" is somewhat misleading as it suggests that you refer to trajectories/air parcels. But what is shown in Figure 4a is the wind.

**Response:** We will change this to "southerly winds around its western flank".

19. L211: Similar to the previous comment. I do not think that you can really see "cold air advection" in this plot. You see northerly winds blowing across a temperature gradient, suggesting cold air advection.

**Response:** We will change this.

20. L207/L214: I was wondering whether the word "distinct" is appropriate

   **Response:** We will change this.

21. here. L214: What is meant by "limited" diabatic cooling?

   **Response:** We will change this to "Some diabatic cooling".

22. L215-216: I suggest mentioning once that the residual is small, e.g., as the last sentence in the section "Lagrangian Temperature Decomposition," and then omitting it in the following text and the figures.

   **Response**: We will add: "The residual is typically small and is thus not further discussed in the following text and the figures"

23. L222: Again, I think the use of the word advection is somewhat misleading here. I suggest to use "southwesterly wind" instead of "southwesterly advection" (se major comment 2).

   **Response:** We will change this.

24. Caption Figure 4: You write "selected grid point" but shown is a "grid box".

   **Response:** We will change this.

25. L249-267: To shorten the entire paragraph: Could you simply say that a DJF cooling event is essentially the same as a "reversed" DJF warming event? To me, Figures 4a and 4d look quite similar, as well as Figures 4b and 4c. And Figure 5k is more or less the same as Figure 5l, just mirrored.

   **Response:** We will streamline and shorten this paragraph, focusing on the reversed behavior between warming and cooling events.

26. L285: I suggest to cross the "which are only briefly discussed", since it gets clear from the "focuses on JJA" that the focus is not on DJF.

   **Response:** We will change this.

27. L333: For instance, at this point it would be very beneficial to be precise of what is meant by cold air advection and warm air advection, i.e. with respect to what is the air cold or warm (see major comment 2).

   **Response:** We will add the meaning of cold air advection and warm air advection to the methodology part.

28. L356/L385/L385/L493: What do you mean with "distinct" patterns and "specific" circulation patterns? Do you mean that all events exhibit a similar pattern, e.g., in the 500 hPa geopotential height? If so, I think you cannot deduce that from the plots, as you only show the mean circulation across all events, which might differ substantially from one event to another.

   **Response:** Yes, we have plotted mean circulation across all events, and event-to-event variability is not quantified here. Nevertheless, if the circulation anomalies between the events were completely different, no consistent mean anomaly would emerge. We will briefly discuss this in the manuscript, also in context with the circulation anomalies shown in the response to reviewer 2 (their major comment 3).

29. L356-373: Are you referring to cooling or warming events here?

**Response:** We are referring to warming events, which will be mentioned in the paragraph in the revised version.

30. L387-388: At this point, I think one must be very careful with the phrasing. You are looking at a budget, so you cannot say that term 1 and term 2 cancel each other out, leaving only term 3 as important! Imagine term 1 being +5 K, term 2 being -5 K, and term 3 being +5 K. You cannot say that term 1 and term 2 cancel each other out, leaving only term 3 as important, nor can you say that term 2 and term 3 cancel each other out, leaving only term 1 as important.

**Response:** You are right, and we will rewrite the paragraph accordingly.

31. L416-425: Again, to shorten the paragraph: Could you simply say that DJF cooling events are essentially the same as a "reversed" DJF warming event?

**Response:** We will change this.

32. L393: I suggest to write "… **presumably** contributing to larger diabatic heating and higher temperatures"

**Response:** We will change this.

33. L500: To better connect to the first part of the sentence, you could insert something like "while the contributions of adiabatic and diabatic processes are generally smaller and vary more in space and also between warm and cold events".

**Response:** We will change this

34. L513/514: Here, scientific debate is still going on. There are also studies saying that heat waves in the mid-latitudes are driven by advection (e.g., Harpaz et al. 2014, Sousa et al. 2019, partly Röthlisberger and Papritz 2023).

**Response:** We will add a more nuanced discussion here.

35. L537: I do not understand what is meant by the sentence "is approximated through the average temperatures of the trajectories initiated on the corresponding day at their initiation time".

**Response:** We will add the following explanation: "This is an approximation, as the trajectories are initialized from different heights above the surface, assuming (and sampling) a well-mixed near-surface layer."

**Technical correction**

36. Title: temperature without plural?

**Response:** We will change this

37. L120: 5th and 95th percentiles of the DTDT change distribution

**Response:** We will change this

38. L125, 136: no plural, DTDT change event?

**Response:** We will change this

39. L190: DTDT variation

    **Response:** We will change this

40. L193: DTDT variation events

    **Response:** We will change this

41. L194: DTDT variations

    **Response:** We will change this

42. Figures: I think it would be helpful to keep the direction of panel labeling consistent across all figures.

    **Response:** We will keep all the panel labels in the vertical direction.

43. L202/284/353/…: Should we invest in the mechanisms?

    **Response:** We will change this

44. L220: On the days of the ...

    **Response:** We will change this

45. L241: smaller in magnitude?

    **Response:** We will change this

46. L339: compared to

    **Response:** We will change this

47. L445: result from

    **Response:** We will change this

**References**

Bieli, M., Pfahl, S., and Wernli, H.: A Lagrangian investigation of hot and cold temperature extremes in Europe, Quarterly Journal of the Royal Meteorological Society, 141, 98-108, 2015.

Röthlisberger, M. and Papritz, L.: Quantifying the physical processes leading to atmospheric hot extremes at a global scale, Nature Geoscience, 16, 210-216, 2023.

---

## Author Comment (AC2)

**Review of "Physical Processes Leading to Extreme day-to-day Temperatures Changes, Part I: Present-day Climate"**

The work of Hamal and Pfhal investigates the physical processes of extremes in the day-to-day variability of near-surface temperature (DTDT) using the ERA5 data set. They use both composites maps and Lagrangian backward trajectories tracking to quantify the mechanisms for these extreme changes in temperature. Their main conclusion is that these extremes are mainly driven by changes in advection in the extra-tropics while the situation is more balanced in the tropics with important contributions from diabatic processes.

The paper is well written and the research question is clearly stated and interesting. The analyses carried out in the paper are careful, well explained and scientifically sound. I should say that the results are not particularly jaw-dropping – I would have guessed that advection was the main contributor of extremes of DTDT in the extra-tropics – but they are nonetheless interesting for documenting these mechanisms. I have some suggestions to improve the quality of the communications of the results and some additional analyses. Therefore I recommend major revisions at this stage, see below for my comments.

We would like to thank the reviewer for their helpful comments. Our responses are printed in blue, whereas the reviewer questions are in black. In addition to addressing the individual comments, we will review the manuscript for clarity and flow.

**Major comments**

1. **Use of ERA5 and HadGHCND:** I think the comparison between the DTDT variability in ERA5 and HadGHCND is problematic. As the authors show in their Figure 1, there are large differences between the two data sets that are probably not physical. The reason is likely because HadGHCND interpolates station data to construct a gridded data set which likely smooths out the daily variability (rather than lack of station coverage I think). In my opinion, this makes the HadGHCND data set particularly not suited for the study that you are doing here. That being said, it is true that the authors mainly compare the spatial patterns rather than the absolute values of sigma_DTDT between the two data sets. If you really want to compare the absolute values found with ERA5 with measured data you should probably go directly to station data. For these reasons, I would discourage to show the comparison with HadGHCND in the figures of the main text: the authors can include it in supplementary materials if they really want to do this comparison. In this case they should also discuss the differences between the two data sets more. Moreover, the rest of the paper does not use HadGHCND.

   **Response:** We agree with the reviewer that the HadGHCND dataset may have smoothed out the variability due to spatial interpolation and the limited number of stations. This can be verified by comparing the HadGHCND dataset with the Berkeley Earth Surface Temperature (BEST) dataset (Figure R1a-h), which incorporates additional data sources beyond HadGHCND. This comparison shows an increased variability pattern in the northern hemisphere for both DJF and JJA (Figures R1b and d). Furthermore, this allows for a more robust comparison with the ERA5 data for all the quantities (Figure R2).

[Figure]

**Figure R1. (a-d) Standard deviation of DTDT variations ($\sigma_{DTDT}$, °C), (e-h)) standard deviation of daily mean temperature ($\sigma_T$, °C), and (i-l) lag-1 autocorrelation of daily mean temperature ($r_{1, T}$) in December-February (DJF) and June-August (JJA) derived from the HadGHCND and BEST datasets.**

Following the reviewer's suggestion, we will use only ERA5 data for the main manuscript, while this observational analysis will be moved to the supplementary material.

[Figure]

**Figure R2. (a, b) Standard deviation of DTDT variations ($\sigma_{DTDT}$, °C), (c, d) standard deviation of daily mean temperature ($\sigma_T$, °C), and (e, f) lag-1 autocorrelation of daily mean temperature ($r_{1, T}$) in December-February (DJF, 1st column) and June-August (JJA, 2nd column) derived from the ERA5 datasets.**

2. **Statistical suggestions:** a. I think it would be interesting to show (at least for the grid points studied) the distribution of delta_T and the quantiles that you are selecting. In particular it would be interesting to see whether the distribution is symmetric. You could for example compute, in addition t its standard deviation, its kurtosis and show the corresponding map.

**Response:** Thank you for your suggestions. We have calculated the DTDT distribution for each selected location during DJF and JJA, as shown in Figure R3. In DJF, North America exhibits the highest variability with a broad distribution, while South America shows the lowest variability with a sharper peak. Europe and Australia display moderate variability, with intermediate kurtosis values and slight distribution asymmetry. However, in JJA, South America becomes more variable, while North America, Europe, and Australia maintain relatively stable distributions with lower variability compared to DJF. Additionally, the distributions become more negatively skewed in JJA. We will add this result to the supplementary material.

[Figure]

**Figure R3. Day-to-day temperature (DTDT) distribution curves over the selected regions: North America (black), Europe (orange), Australia (green), and South America (purple) for (a) December-February (DJF) and (b) June-August (JJA). The small dots on the left and right represent the 5th and 95th percentiles, respectively.**

b. One question I had while reading the paper is how much the extremes of delta_T relate to extremes of T, in other words: do your warming/cooling events also correspond to warm/cool extremes? I think it would be super interesting to show how the extremes of delta_T are linked to the quantiles of $T_t$ and $T_{t-1}$. For example, do extreme warming events happen because we start from a very cold quantile and we end up in the middle of the temperature distribution or do we start from the middle of the distribution and end up in the right tail? The physical processes in these two situations are likely different.

**Response:** Thank you for your insightful suggestion. We have illustrated the relationship between DTDT changes and the specific quantiles (terciles) of $T_t$ and $T_{t-1}$ in Figure R4. Our analysis reveals that extreme warming events originate in the lower to middle-temperature quantiles of $T_{t-1}$ and shift toward the middle to higher quantiles of $T_t$. Conversely, extreme cooling events typically begin in the middle to higher quantiles of $T_{t-1}$ and shift to the middle to lower quantiles of $T_t$. We will incorporate this analysis into the manuscript in Section 3.2.

[Figure]

**Figure R4. Heatmaps of the relationship between DTDT change and the deciles of temperature on the previous day ($T_{t-1}$) and the event day ($T_t$) for December-February (DJF) and (b) June-August (JJA) for North America. The x-axis and y-axis represent deciles of $T_t$ and $T_{t-1}$, while the color shading indicates DTDT changes, with red and blue colors indicating warming and cooling, respectively. The black circles represent extreme DTDT changes.**

3.  **Comparison with climatology:** I find it really interesting that in Figure 4-5 and others the warming and cooling events seem the reverse of one another. As advection seems to be the largest contributor, it seems to me that extremes of sigma_DTDT happens as if this mechanism was switched on or off: warming events happen because the northward advection was switched off and vice versa for cold events. This leads me to my question which is not unrelated to my previous comment 2.a., how are the dynamical situations that you identify unusual with respect to climatology? Is it the starting point that is dynamically unusual or the end point? To be more clear, it seems to me that you should probably do your composite maps also in anomalies.

    **Response:** Thank you for your insightful suggestion. We have analyzed the atmospheric circulation anomalies for the two days involved in an extreme DTDT change event with respect to the seasonal climatology, revealing significant deviations from the mean. Specifically, warming events are associated with southerly wind anomalies and higher geopotential heights (Figure R5), while cooling events are linked to northerly wind anomalies and lower geopotential heights. We will add this to the supplementary.

[Figure]

**Figure R5. Composite of near-surface temperature anomalies (T2M, °C, color shading), wind anomalies at 850 hPa (UV, m/s, vectors), and geopotential height anomalies at 500 hPa (GP, gpm, magenta contours, dotted and bold magenta contours show negative and positive values, respectively) with respect to seasonal mean on the (a, c) previous day (t-1) and (b, d) event day (t) of the warming (a,b) and cooling events (c,d) during December-February (DJF) at a selected grid in North America (red grid). Note that wind vector anomalies ≥2m/s are plotted.**

4. **Extremes of DTDT and fronts:** the fact that advection is the main factor of extremes in DTDT in the extra-tropics is not super surprising and that is what I would expect because of the existence of atmospheric fronts (some may even argue that fronts are by definition extremes of DTDT). I am surprised that the authors do not mention at all these structures. Can you say a word about how your analysis and results relate to the literature on frontal structures? Moreover, it seems to me that frontal structures are well identified in the climate variability literature as being the mechanism for day-to-day variability (e.g. Ghil and Lucarini (2020)), maybe you could also mention that in the broader context of your work.

**Response:** We agree that atmospheric fronts play a pivotal role in shaping DTDT extremes in the extratropics. Baroclinic instability drives the formation of frontal structures, which are closely linked to the development of cyclones and anticyclones and serve as key drivers of day-to-day temperature variability (Ghil and Lucarini, 2020). In the composite anomalies of warming events, the transition from cold air masses on the preceding day to warm air masses on the event day corresponds to the passage of warm fronts, which are associated with strong spatial temperature gradients (Figure R5a-b). This phenomenon has been extensively studied and confirmed for European DTDT extremes using different frontal structures (cold, warm, and occluded) (Piskala and Huth, 2020). Our primary objective was to identify the dominant processes driving DTDT extremes. Since our study did not include a database of frontal passages, we initially did not reference this aspect in our analysis. However, we will now incorporate a discussion of this topic, including references to the papers mentioned above, in both the introduction and discussion sections.

**Minor comments**

1. Paragraph L49: in this paragraph you are mainly talking about hot and cold extremes, which are rather different from the extremes in DTDT that you are looking at. This may be confusing for the reader, please be more clear about how the extremes per se relate to the extremes of DTDT (see also my major comment 2).

   **Response:** We agree that extreme daily temperature events and extreme DTDT change events are two different things. However, there is limited literature on the variability of DTDT extremes; we introduce daily temperature extremes to provide the context for atmospheric circulation in the introduction. Furthermore, Equation 5 and Figure R4 illustrate how DTDT variability relates to daily temperature variability. Additionally, our analysis of composites of extreme DTDT and daily extreme events reveals similar circulation patterns, which are, however, more pronounced in the case of daily temperature extremes shown in the response to reviewer 1 (their major comment 1).

2. For clarity, I think you should detail a bit more the terms in eq 1-5. In particular, equation 4 is not necessary to me and may be confusing. Moreover, you should explain what the approximation means in equation 3 (explain why this is actually a very good approximation and the errors involved are small because of the typical time scale of the seasonal cycle).

   **Response:** We will explain equations 1-5 in more detail:

   This study defines DTDT change, denoted as $\delta_T$, as the difference in daily mean near-surface air temperature between the previous day ($T_{t-1}$) and the day of the event ($T_t$), as shown in Eq. (1).

   $$\delta_T = (T_t - T_{t-1}) \qquad (1)$$

   The average daily temperature change, $\mu_{DTDT}$ reflects the difference between the temperatures at the start ($T_0$) and end ($T_n$) of the time series (Eq. 2).

   $$\mu_{DTDT} = \frac{1}{n}\sum_{t=1}^{n}(T_t - T_{t-1}) = T_n - T_0 \qquad (2)$$

   To capture typical day-to-day temperature changes, we thus use the standard deviation, $\sigma_{DTDT}$, as shown in Eq. (3).

   $$\sigma_{DTDT}{}^2 = \frac{1}{n}\sum_{t=1}^{n}(T_t - T_{t-1})^2 \qquad (3)$$

   By inserting the average daily temperature $\mu_T$ and multiplying out the square bracket, we find a relationship between $\sigma_{DTDT}$, the standard deviation of the daily mean temperature ($\sigma_T$) and the covariance between consecutive days ($COV(T_t, T_{t-1})$):

   $$\sigma_{DTDT}{}^2 = \frac{1}{n}\sum_{t=1}^{n}((T_t - \mu_T) - (T_{t-1} - \mu_T))^2$$

   $$= \frac{1}{n}\sum_{t=1}^{n}((T_t - \mu_T)^2 + (T_{t-1} - \mu_T)^2 - 2(T_t - \mu_T)(T_{t-1} - \mu_T)$$

   $$\approx 2\sigma_T^2 - 2COV(T_t, T_{t-1}) \qquad (4)$$

   The approximation in equation (4) is associated with the fact that, for large n, both $\frac{1}{n}\sum_{t=1}^{n}(T_{t-1} - \mu_T))^2$ and $\frac{1}{n}\sum_{t=1}^{n}(T_t - \mu_T))^2$ are good estimators of $\sigma_T^2$. Finally, the standard deviation of DTDT can thus be expressed as a function of the usual standard deviation ($\sigma_T$) and the lag-1 autocorrelation $r_{1,T}$ of daily mean temperature, as shown in Eq. (5).

$$\sigma_{DTDT} = \sigma_T \sqrt{2\,(1\text{-}r_{1,T}\,)} \qquad\qquad (5)$$

3. L126: can you detail a bit more why those choices were made, especially the date and time of the initialization of the backward trajectory.

   **Response:** We will add a more detailed explanation:

   The Lagrangian analysis tool (LAGRANTO), introduced by Sprenger and Wernli (2015), is used to calculate backward trajectories of near-surface air masses on days associated with extreme DTDT changes from 1980 to 2020. The trajectories are initialized at 18 UTC on both the preceding day (t-1) and on the event day (t) at 10, 30, 50, and 100 hPa above the surface at the corresponding grid cells. Similar to previous studies on extreme temperatures (Zschenderlein et al., 2019), the different initialization heights are used to sample a near-surface layer that is assumed to be well-mixed. The time difference of 24 hours between the two initializations allows for a proper separation of the air masses before and after the temperature change. Although we use LAGRANTO to calculate 10-day backward trajectories, extremes typically develop on a timescale of 2–3 days (Bieli et al., 2015; Röthlisberger and Papritz, 2023). Therefore, we focus on 3-day backward trajectories for our analysis. Various variables of interest, including latitude, longitude, pressure, temperature, and potential temperature, are interpolated along the trajectory paths and saved at 1-hour intervals.

4. Equation 6: this is more for my understanding: given that you are looking at air parcels close to the ground, how can the adiabatic contribution be anything else than positive?

   **Response:** While the air masses always undergo adiabatic warming (since they arrive near the surface), the magnitude of this warming can be different, with some air masses descending more than others. Accordingly, the contribution to DTDT changes can be both negative or positive, depending on typical differences in the strength of the descent between the two days. Our results indicate that the mean effect of such changes in adiabatic warming is relatively small in many regions, but they contribute substantially to event-to-event variability.

5. Figure 1: the colors scale in all panels is unfortunate. You are showing only positive, non-divergent values therefore you should not use a divergent color maps which is misleading for the reader. Also, because you compare between panels a,b and c,d, the values of the color map should have the same range. Finally, you should probably use the Robinson projection.

   **Response:** Thank you for your suggestion. We experimented with a non-divergent color bar but found that the differences in the spatial distribution of DTDT were less clear. To maintain clarity and consistency, we will use the same color bar across all figures, with a similar range for direct comparison. Additionally, we will apply the Robinson projection to improve the representation of spatial patterns (e.g., Figure R1).

6. L149 and following: You mention the "magnitude of sigma_DTDT changes." I am not sure what this refers to. If I understand correctly, you should rather talk about the "magnitude of sigma_DTDT."

   **Response:** Thank you for the suggestion. We will change this to "magnitude of $\sigma_{DTDT}$".

7. L174: what are "the deep tropics" ?

   **Response:** Here, we mean to indicate the core equatorial region.

8. Because you are studying land grid points only and their proportion varies a lot between the latitudes, I am not sure the zonal means in Figure 1 and following are really relevant: the reader can

see by themselves that there is a marked latitudinal gradient of the quantities you are displaying. Also, do you have an explanation for why in the southern hemisphere the variability is much smaller than in the northern hemisphere for the grid points with the same absolute latitude?

**Response:** Thank you for the suggestion. We will remove the zonal mean representation (e.g., Figure R1).

While, in general, differences in the land-ocean distribution and corresponding spatial temperature gradients may lead to different magnitudes of the variability between the hemispheres (via advection), we do not think that the variability in the southern hemisphere is much smaller than in the Northern Hemisphere at the same latitude. The fact that the maps have shown the latitude range from 60°S to 90°N might make the comparison a bit difficult before. However, we have now extended maps to 90°S and 90°N, and a larger magnitude of DTDT is clearly observed in high latitudes, such as Antarctica (Figure R2).

The variability in the Southern Hemisphere is generally lower than in the Northern Hemisphere at the same latitude in DJF. As shown earlier, advection is the key driver of this variability, and the magnitude is thus related to horizontal temperature gradients. In the Northern Hemisphere, larger land masses are associated with larger temperature contrasts between continents and ocean, while in the Southern Hemisphere, in particular at latitudes south of 40°S, such land-sea differences are much smaller due to the small land fraction and the dominance of oceanic air masses. During JJA, when the meridional temperature gradient is larger in the Southern Hemisphere, the magnitude of the variability becomes more comparable between the two hemispheres.

9.  L205: "DJF warm events": I would strongly discourage you from using this phrasing for the events you are studying because it is really misleading for the reader. You should talk about warming/cooling events.

    **Response:** We will change "warm/cold events" to "warming/cooling events".

10. Fig4: a. Please define more precisely near surface temperature: is it T2M? b. I would suggest not to use absolute values for composite maps: first because there is still the seasonal cycle (how do you handle that by the way?) and second because it is difficult to read and the sudden change of temperature from t-1 to t is not very clear. Maybe simply use anomaly maps and/or make a difference between the map at t and at t-1?

    **Response:** Yes, near-surface temperature (T2M) is indeed used, as mentioned in the methodology section. In response to the reviewer's suggestion, we have plotted composite maps for the two days involved in an extreme DTDT change event relative to the seasonal climatology, highlighting significant deviations from the mean (Figure R5). We appreciate the reviewer's recommendation, as the sudden temperature change from t-1 to t is now clearly evident.

    Additionally, we plotted the difference between the event day and the previous day, along with the absolute values, to illustrate both the changes and their magnitudes (Figure R6). In the revised manuscript, we will present the circulation patterns on each day along with their differences, while the climatology anomalies will be included in the supplementary material.

[Figure]

**Figure R6. Composite of near-surface temperature (T2M, °C, color shading), wind at 850 hPa (UV, m/s, vectors), and geopotential height at 500 hPa (GP, gpm, magenta contours) on the (a, c) previous day (t-1), (b, d) event day (t ) and (c, f) difference of event day and previous day of the warming (a-c) and cooling (d-f) events during December-February (DJF) at a selected grid in North America (red grid). Note that (a-d) wind vectors ≥5m/s and (e-f) wind anomalies ≥1m/s are plotted. The dotted and bold magenta contour in c and f indicate negative and positive geopotential height differences, respectively.**

11. Figure 5 and alike: it would be clearer for the reader if you could indicate explicitly on the figure (not only in the legend) if those are DJF or JJA events**.**

    **Response:** Thank you for the suggestion. We now indicate whether DJF or JJA events are shown in the Figures themselves (e.g. Figure R6).

12. L431: "To systematically investigate the mechanism driving DTDT extremes over the subtropics in the southern hemisphere during DJF and JJA, we select a specific location in Australia": to me this sentence sounds self-contradictory, how can you systematically investigate if you look at only one grid point?

**Response:** We have improved this sentence structure as "To investigate the mechanism driving extremes DTDT events over the subtropics in the southern hemisphere during DJF and JJA, we select a specific location in Australia."

13. The conclusions reached are based on the analysis of only some grid points at various longitudes/latitudes. Although I think the conclusions reached can probably be extended to the other grid points in the vicinity, I think the authors should be a bit more cautious in their concluding statements.

**Response:** The results for a few additional grid points (Northern Asia, Southern South America, South Asia, Africa, and Western North America) are presented in the supplementary material. Nevertheless, we will revise the wording to make it more cautious regarding potential spatial variability.

14. L531: "This dominant effect of advection also explains why the magnitude of DTDT changes is typically larger in the extratropics, where horizontal temperature gradients and wind velocities are larger compared to the tropics": this statement is likely true but deserves more evidence.

**Response:** Here, we are not sure which kind of evidence the reviewer would like to see. Based on our composite analysis of atmospheric circulation and 3d backward trajectories, we show that advection dominates DTDT changes in the extratropics. The fact that horizontal temperature gradients and wind velocities are larger in the extratropics than in the tropics is evident from basic climatological data. The relationship between advection, wind velocity, and temperature gradient is clear from the Eulerian version of the thermodynamic equation, where the advection term is written as the scalar product of the horizontal wind vector and temperature gradient. Finally, the magnitude of the adiabatic and diabatic terms in our Lagrangian budgets are of the same magnitude in the extratropics and tropics and thus cannot compensate for the difference in advection. A note on the last point will be added to the conclusion section.

**References**

Ghil, M., & Lucarini, V. (2020). The physics of climate variability and climate change. *Reviews of Modern Physics*, *92*(3), 035002.

Bieli, M., Pfahl, S., and Wernli, H.: A Lagrangian investigation of hot and cold temperature extremes in Europe, Quarterly Journal of the Royal Meteorological Society, 141, 98-108, 2015.

Piskala, V. and Huth, R.: Asymmetry of day-to-day temperature changes and its causes, Theoretical and Applied Climatology, 140, 683-690, 2020.

Röthlisberger, M. and Papritz, L.: Quantifying the physical processes leading to atmospheric hot extremes at a global scale, Nature Geoscience, 16, 210-216, 2023.

Zschenderlein, P., Fink, A. H., Pfahl, S., and Wernli, H.: Processes determining heat waves across different European climates, Quarterly Journal of the Royal Meteorological Society, 145, 2973-2989, 2019.

---

## Referee Report (RR1)

**Second round of review of**
**"Physical Processes Leading to Extreme day-to-day Temperatures Changes, Part 1: Present-day Climate"**
**by Kalpana Hamal and Stephan Pfahl**
**submitted to Weather and Climate Dynamics**

**General comment:**
This is the second time I am reviewing the manuscript, and I believe it has substantially improved during the first round of review. In particular, I appreciate that the authors are now more precise in their use of terminology (e.g., "advection," warming/cooling events), which significantly enhances the clarity of the text.

However, in response to some of my previous comments (regarding lines 159-161, twice line 214, Caption Figure 4 in the original manuscript), the authors stated that they had incorporated the suggested changes into the revised manuscript. Unfortunately, these changes do not appear in the current version. I assume this was an oversight, and I would like to encourage the authors to implement these revisions in the next version.

I still have a few minor suggestions for improving the text, but overall, I feel the manuscript is close to being ready for publication.

**Minor comments:**

L61: The sentence "In contrast, tropical regions typically exhibit weaker temperature advection" suggests a comparison, but earlier in the text, you have not explicitly mentioned that other regions exhibit stronger temperature advection. Consider rephrasing to make the comparison clearer and more logically connected.

L61: I feel that the use of "However" at this point may not be appropriate.

L112: "The approximation in equation (4) is based on ..." instead of "… is associated with"?

L124: I suggest removing the word "previous" from the phrase "previous studies on extreme temperatures" as it may imply that your study also focuses on extreme temperatures, which it does not.

L125: It is not clear to me why the near-surface layer must be assumed to be well-mixed. Could you clarify this point?

L127-129: Think about just omitting the fact that you actually computed 10 day trajectories, although in the end you only needed 3 day trajectories.

L136: Where does the "these" refer to?

L137: I would try to be consistent with the heading of this subsection, so I suggest instead of "Lagrangian temperature variation decomposition" "Lagrangian temperature variability decomposition".

L186: "… while in the tropics, $\sigma_{DTDT}$ is lower associated with lower $\sigma_T$, despite lower $r_{1,T}$." I have difficulties to understand this sentence. Think about reprhasing.

L541-544: "… but advection plays a smaller role, in particular for temperature extremes and heat waves in larger parts of the mid-latitudes". I appreciate that you tried to add a more nuanced discussion here. However, I still feel that it is not correct what is stated here, since the literature is not clear about whether advection really plays a smaller role for warm extremes than for cold extremes. Maybe just apply a more cautios formulation, e.g. "… where advection is sometimes thought to play a smaller role, in particular for temperature extremes and heat waves in larger parts of the mid-latitudes"?

L576: Where does the "this" refer to?

**Technical corrections:**

L147: cross the "was"?

---

## Author Response (AR3)

**First round of review of "Physical Processes Leading to Extreme day-to-day Temperatures Changes, Part I: Present-day Climate" by Kalpana Hamal and Stephan Pfahl submitted to Weather and Climate Dynamics.**

We want to thank the reviewers for their helpful comments, which helped to improve our manuscript. This document is structured as follows: We address all comments of reviewers 1 and 2, respectively. Our responses are in blue, whereas the reviewer's questions are in black. In addition to addressing the individual comments, we have reviewed the manuscript for clarity and flow.

**Comments from Reviewer 1 and Responses**

**1.   Major comments- Reviewer 1**

a.  To me, it is often unclear whether the text is addressing temperature extremes or extreme temperature variability. I believe this is due to imprecise terminology. For example, throughout the text, the terms "warm events" and "cold events" are used to describe extreme day-to-day temperature change events. However, I find these expressions somewhat misleading, as they may imply that you are looking at warm and cold temperature extremes themselves, rather than on extreme warming and cooling events. The text would be much clearer if "warm events" were replaced with "warming events" and "cold events" with "cooling events." The same is true for the expression "DTDT extremes", which I think should be better replaced by something like "extreme DTDT changes". Furthermore, at some points (introduction, summary), you make a connection between your work and studies focusing on the mechanisms behind the development of warm and cold temperature extremes. I think it should be made clearer at this point that an extreme temperature event does not necessarily have to be linked to extreme temperature variability. A sudden temperature increase does not necessarily occur when it gets particularly hot, and similarly, a sharp temperature drop does not have to happen when it is particularly cold.

**Response:** Thank you for your feedback. Throughout the paper, we have now used 'warming events' and 'cooling events' instead of 'warm events' and 'cold events,' which improves clarity. We also rephrased 'DTDT extremes' as 'extreme DTDT changes' to avoid ambiguity.

We appreciate your comment and agree that clarifying the distinction between extreme temperature events and extreme DTDT changes is important. While we acknowledge that extreme temperature events and extreme DTDT changes are not necessarily linked, our focus on daily temperature extremes in the introduction provides context for atmospheric circulation patterns, particularly given the limited literature on extreme DTDT variability.

Our analysis, as also indicated by Equation (5), shows that variations in $\sigma_T$ primarily determine the spatial variability of $\sigma_{DTDT}$. Additionally, we explore the relationship between DTDT changes and specific quantiles (deciles) of $T_t$ and $T_{t-1}$, as illustrated in Figure R1. Extreme warming events are associated with changes from lower-middle temperature quantiles ($T_{t-1}$) to middle-higher temperature quantiles ($T_t$), while the reverse is true for cooling events. Thus, extreme DTDT changes typically occur when $T_t$ predominantly falls within the tail quantiles (stippling in Figure R1).

$$\sigma_{DTDT} = \sigma_T \sqrt{2\,(1\text{-}r_{1,T})} \qquad\qquad (5)$$

[Figure]

**Figure R1. Heatmaps of the relationship between DTDT change and the deciles of temperature on the previous day (T$_{t-1}$) and the event day (T$_t$) for (a) December-February (DJF) and (b) June-August (JJA) for North America. The x-axis and y-axis represent deciles of T$_t$ and T$_{t-1}$, while the color shading indicates DTDT changes, with red and blue colors indicating warming and cooling, respectively. The black circles represent extreme DTDT changes.**

To further illustrate the differences in atmospheric circulation between extreme DTDT changes and daily extreme events, for example, the corresponding composites for the winter season and the selected location in North America are shown in Figure R2. They reveal similar circulation patterns, which, however, are more pronounced in the case of daily temperature extremes (Figure R2). Similar results are observed at other grid points. We have added this analysis as a supplementary in Figure S7 (But for the location Europe).

[Figure]

**Figure R2. Composite of near-surface temperature (T2M, °C, color shading), wind at 850 hPa (UV, m/s, vectors), and Geopotential height at 500 hPa (GP, gpm, magenta contours) on the (a, c) previous day (t-1) and (b, d) event day (t ) of warming (a,b) and cooling events (c,d) during December-February (DJF) at a selected grid point in North America (red box). Composites of (e) warm and (f) cold events, defined with the 5th and 95th percentiles of the daily temperature distribution as thresholds, are shown in the third row, and the difference between daily extremes (panels e and f) and DTDT extremes (panels c and d) are shown in panels g-h. Note that in a-f wind vectors ≥5m/s and in g-h wind vector anomalies ≥1m/s are plotted. The dotted and bold magenta contours in g-h indicate negative and positive geopotential values, respectively.**

b. Overall, I feel that the term "advection" is not applied with sufficient precision. At times, it is used when showing instantaneous wind fields (e.g. L210, L2022), and at other times when discussing trajectories and the transport of air masses (e.g. L211, L217). I believe it is crucial to be very careful about when the term "advection" is used and when it might be more appropriate to use a different term, especially considering that the existing literature is not always clear on this matter. Furthermore, I think it would be very helpful to clarify what is meant by "cold air advection" and "warm air advection," i.e. with respect to what is the air cold or warm. I believe what you mean is that, for instance, the air transported to the location on the day of a cooling event is originally colder than the air that was transported there the previous day. I think it would be very beneficial to be more precise here.

**Response:** Thank you for the suggestion. We have updated the usage of the term advection and clarified the cold and warm advection terms in the methodology and Appendix. To avoid confusion, the term is used exclusively in the discussion of air mass transport (trajectories) and not anymore for wind composites.

Technically, the advection term in our Lagrangian budget is defined as follows: As most temperature extremes were found to develop within a 2–3-day timescale (Bieli et al., 2015), we selected 3-day backward trajectories for the budget calculations in our study. $\overline{T}_t^{-3d}$ represents the average temperature of the air parcels initialized on the day of the extreme event, three days before their arrival at the target location, while $\overline{T}_{t-1}^{-3d}$ represents the corresponding temperature for the air parcels initialized one day earlier. The expression $\delta_T^{-3d} = \overline{T}_t^{-3d} - \overline{T}_{t-1}^{-3d}$ Thus, it captures the difference in temperature between the air parcels three days before their arrival. Assuming no further temperature changes occurred during transport, the DTDT change is solely due to these initial differences. This suggests that variations in the advection of air parcels with different original temperatures between the previous day and the day of the event cause the temperature changes. This is why we refer to this term as an advection term. When the temperature of the air parcels was initialized on the previous day ($\overline{T}_{t-1}^{-3d}$) is higher than the temperature of the air parcels at the event ($\overline{T}_t^{-3d}$), this represents a shift from warmer air to colder air on the event day, which is referred to as cold air advection. The reverse is true for warm air advection.

c. My last major comment relates to adiabatic warming. Whenever you describe the temperature at t-1, you write something similar to L216: "Accordingly, the temperature at t-1 is mainly determined by cold air advection, mitigated by adiabatic warming, ...". I was wondering if it is not always the case that an air mass at the surface undergoes adiabatic warming. If so, is it truly necessary—or perhaps redundant—to specify each time that the temperature has been affected by some adiabatic warming? The same applies to the initial temperature, which is always lower than the final temperature. I believe the text could be condensed on this point such that greater emphasis on the differences between day t-1 and day t is given.

**Response:** Thank you for the suggestion. Our approach involves studying the temperature evolution on each of the two days individually, followed by analyzing their differences. While the air masses always undergo adiabatic warming (since they arrive near the surface), the magnitude of this warming can be different, with some air masses descending more than others. Accordingly, the contribution to DTDT changes can be both negative or positive, depending on typical differences in the strength of the descent between the two days. Our results indicate that the mean effect of such changes in adiabatic warming is relatively small in many regions, but they contribute substantially to event-to-event variability.

Nevertheless, we understand the reviewer's point that the text should mainly focus on explaining the differences between the two days that make up the DTDT changes, and we have thus further condensed the descriptions of the evolution during the individual days.

**2. Minor comments- Reviewer 1**

2.1 L102-117: This paragraph is very hard to follow. I think this is because the expressions used in the formulas are not well described and because equations (3) and (4) are never explained in the text. I suggest to break down the equations to what is really important and introducing the equations step by step, instead of a full block of equations.

**Response:** We have explained equations 1-5 in more detail in section 2.2:

This study defines DTDT change, denoted as $\delta_T$, as the difference in daily mean near-surface air temperature between the previous day ($T_{t-1}$) and the day of the event ($T_t$), as shown in Eq. (1).

$$\delta_T = (T_t - T_{t-1})$$  (1)

The average daily temperature change, $\mu_{DTDT}$ reflects the difference between the temperatures at the start $(T_0)$ and end $(T_n)$ of the time series (Eq. 2).

$$\mu_{DTDT} = \frac{1}{n}\sum_{t=1}^{n}(T_t - T_{t-1}) = T_n - T_0$$  (2)

To capture typical day-to-day temperature changes, we thus use the standard deviation, $\sigma_{DTDT}$, as shown in Eq. (3).

$$\sigma_{DTDT}^2 = \frac{1}{n}\sum_{t=1}^{n}(T_t - T_{t-1})^2$$  (3)

By inserting the average daily temperature $\mu_T$ and multiplying out the square bracket, we find a relationship between $\sigma_{DTDT}$, the standard deviation of the daily mean temperature $(\sigma_T)$ and the covariance between consecutive days (COV $(T_t, T_{t-1})$):

$$\sigma_{DTDT}^2 = \frac{1}{n}\sum_{t=1}^{n}((T_t - \mu_T) - (T_{t-1} - \mu_T))^2$$
$$= \frac{1}{n}\sum_{t=1}^{n}((T_t - \mu_T)^2 + (T_{t-1} - \mu_T)^2 - 2(T_t - \mu_T)(T_{t-1} - \mu_T)$$

$$\approx 2\sigma_T^2 - 2\text{COV}(T_t, T_{t-1})$$  (4)

The approximation in equation (4) is associated with the fact that, for large n, both $\frac{1}{n}\sum_{t=1}^{n}(T_{t-1} - \mu_T))^2$ and $\frac{1}{n}\sum_{t=1}^{n}(T_t - \mu_T))^2$ are good estimators of $\sigma_T^2$. Finally, the standard deviation of DTDT can thus be expressed as a function of the usual standard deviation $(\sigma_T)$ and the lag-1 autocorrelation $r_{1,T}$ of daily mean temperature, as shown in Eq. (5).

$$\sigma_{DTDT} = \sigma_T \sqrt{2(1 - r_{1,T})}$$  (5)

2.2 L121-122: The number of identified events should differ between the ERA5 data and the HadGHCND data, since their used periods differ. To which dataset do the numbers refer to?

Response: Yes, the number of events differs depending on the choice of datasets. In Figures 1 and Figure S1, we used Best and HadGHCND data to compare DTDT variability patterns with ERA5. For extreme DTDT events, we only used the ERA5 dataset, as shown in Figure 2 (in the paper). The sentence has been updated accordingly in the paper.

2.3 L131: To better understand the underlying mechanism of what?

Response: We have added the following: "To better understand the underlying mechanisms of extreme DTDT changes."

2.4 L133: Instead of "apply a novel" "introduce a novel" to make clear that this precise decomposition has not been used before?

Response: We have changed this.

2.5 L133: Lagrangian temperature variability decomposition?

Response: We have changed this.

2.6 L137: The phrase "The Lagrangian decomposition of DTDT changes, as approximated by the trajectories" is odd, since the trajectories do not approximate the Lagrangian decomposition. Rather, the Lagrangian decomposition is obtained from computed trajectories.

Response: We have changed this.

2.7  L 141: What is decomposed?

**Response:** We have improved the sentence as "The DTDT change has been decomposed into three contributing factors."

2.8  L142: I think at this point it would be very worth noting that advection in this approach refers to something different than in the approach by Röthlisberger and Papritz (relates to major comment 2).

**Response:** Yes, we have used a different method to calculate the advection term than Röthlisberger and Papritz (2023). For daily temperature extremes, they use horizontal advection of the air parcel in the direction of the climatological temperature gradient. As we did not decompose the temperature into anomalies and climatology (which is less useful for the analysis of DTDT changes, for which the temperature is not necessarily very anomalous on both involved days), we have used a simpler definition, as explained in our response to the second major comment. This explanation has been added to the revised manuscript.

2.9  L149: instead of "magnitude of σ changes," either "magnitude of DTDT variations, quantified by σ", or simply "magnitude of σ"

**Response:** We have improved the sentence as "Both the Observation and ERA5 datasets reveal that the magnitude of DTDT variations, quantified by $\sigma_{DTDT}$ is larger…"

2.10  L150-153: Here it is written "the variability is larger during DJF than in JJA", followed by "the variability is above 3 °C in DJF compared to 1-4 °C in JJA". However, 4 °C is larger than 3 °C, such that one could conclude that the variability in JJA is larger than in DJF. Please rephrase this sentence more precisely.

**Response:** We have rephrased to "Notably, $\sigma_{DTDT}$ is larger during DJF than in JJA in the Northern Hemisphere, while the seasonal cycle is less clear in the Southern Hemisphere, with higher $\sigma_{DTDT}$ in JJA in Antarctica, central South America, South Africa, and northern Australia, but higher $\sigma_{DTDT}$ in DJF in southern Australia".

2.11  L150-153: Here it is shown that the magnitude in DTDT changes is larger in DJF than in JJA, irrespective of the hemisphere. This means that in the northern hemisphere, the magnitude of DTDT changes is larger in winter than in summer. In contrast, in the southern hemisphere, the magnitude of DTDT changes is larger in summer than in winter. Is this behavior expected? Can you think of any explanations for this behavior?

**Response:** We expect this behavior during the northern hemisphere winter, as the atmospheric circulation is stronger (Figure 4 in the paper). As anticipated, both warming and cooling events in the hemisphere are primarily driven by advection. In the southern hemisphere, the seasonality is less clear and depends on the region and dataset. We have mitigated our statement in the manuscript accordingly. For our selected location in southern Australia, the large DTDT changes during the DJF season (summer for this region), are due to a more dominant role of diabatic processes associated with stronger heating of the air masses over the continent due to stronger surface fluxes in summer.

2.12  L153: rephrase "remain consistently"

**Response**: We have rephrased to "However, $\sigma_{DTDT}$ is around 1-2°C in the tropics and 1-3°C over higher-latitude land regions in the southern hemisphere…."

2.13  L153/L155 and other lines: Similar to comment to L149: To my understanding the phrases "σ variations" and "σ changes" do not make sense. I think it would be more accurate to simply use "σ".

**Response:** We have changed this.

2.14  L159-161: "Since the magnitude of σ changes can be expressed as a function of …, Figures 1 and 2 show these related quantities." Again, I think this sentence is not properly formulated. The Figures 1 and 2 do not show the other quantities because σ can be expressed as a function of them. It is rather that you decided to show them as they are part of the computation of σ.

**Response:** The sentence has been reformulated: "According to equation 5, the magnitude of DTDT temperature changes can be expressed as a function of the standard deviation $\sigma_T$, and lag-1 autocorrelation

$r_{1,T}$ of daily mean temperature, which are also shown in Fig. 1 and 2."

2.15 Figure 1/Figure 2: It would be helpful to use the same colorbars in Figure 1 and Figure 2 to enable an easier comparison.

**Response:** We have adapted the same color bar in the revised manuscript for easier comparison.

2.16 L183/185: I think it is incorrect to use the phrase "leads to" here. Replace by something like "associated with".

**Response:** We have changed this.

2.17 L184: "smaller" instead of "lower"?

**Response:** We have changed this.

2.18 L209: I feel that the phrase "southerly airflow around its western flank" is somewhat misleading as it suggests that you refer to trajectories/air parcels. But what is shown in Figure 4a is the wind.

**Response:** We have changed this to "southerly winds around its western flank".

2.19 L211: Similar to the previous comment. I do not think that you can really see "cold air advection" in this plot. You see northerly winds blowing across a temperature gradient, suggesting cold air advection.

**Response:** We have changed this.

2.20 L207/L214: I was wondering whether the word "distinct" is appropriate

**Response:** We have changed this.

2.21 here. L214: What is meant by "limited" diabatic cooling?

**Response:** We have changed this to "Some diabatic cooling."

2.22 L215-216: I suggest mentioning once that the residual is small, e.g., as the last sentence in the section "Lagrangian Temperature Decomposition," and then omitting it in the following text and the figures.

**Response**: We have added: "The residual is typically small and is thus not further discussed in the following text and the figures."

2.23 L222: Again, I think the use of the word advection is somewhat misleading here. I suggest to use "southwesterly wind" instead of "southwesterly advection" (se major comment 2).

**Response:** We have changed this.

2.24 Caption Figure 4: You write "selected grid point" but shown is a "grid box".

**Response:** We have changed this.

2.25 L249-267: To shorten the entire paragraph: Could you simply say that a DJF cooling event is essentially the same as a "reversed" DJF warming event? To me, Figures 4a and 4d look quite similar, as well as Figures 4b and 4c. And Figure 5k is more or less the same as Figure 5l, just mirrored.

**Response:** We have streamlined and shortened this paragraph, focusing on the reversed behavior between warming and cooling events.

2.26 L285: I suggest to cross the "which are only briefly discussed", since it gets clear from the "focuses on JJA" that the focus is not on DJF.

**Response:** We have changed this.

2.27 L333: For instance, at this point it would be very beneficial to be precise of what is meant by cold air advection and warm air advection, i.e. with respect to what is the air cold or warm (see major comment 2).

**Response:** We have added the meaning of cold air advection and warm air advection to the methodology part.

2.28 L356/L385/L385/L493: What do you mean with "distinct" patterns and "specific" circulation patterns? Do you mean that all events exhibit a similar pattern, e.g., in the 500 hPa geopotential height? If so, I think you cannot deduce that from the plots, as you only show the mean circulation across all events, which might differ substantially from one event to another.

**Response:** Yes, we have plotted mean circulation across all events, and event-to-event variability is not quantified here. Nevertheless, if the circulation anomalies between the events were completely different, no consistent mean anomaly would emerge. We have briefly discussed this in the manuscript, also in context with the circulation anomalies shown in the response to reviewer 2 (their major comment 3).

2.29 L356-373: Are you referring to cooling or warming events here?

**Response:** We are referring to warming events, which have been mentioned in the revised version of the paragraph.

2.30 L387-388: At this point, I think one must be very careful with the phrasing. You are looking at a budget, so you cannot say that term 1 and term 2 cancel each other out, leaving only term 3 as important! Imagine term 1 being +5 K, term 2 being -5 K, and term 3 being +5 K. You cannot say that term 1 and term 2 cancel each other out, leaving only term 3 as important, nor can you say that term 2 and term 3 cancel each other out, leaving only term 1 as important.

**Response:** You are right, and we have rewritten the paragraph accordingly.

2.31 L416-425: Again, to shorten the paragraph: Could you simply say that DJF cooling events are essentially the same as a "reversed" DJF warming event?

**Response:** We have changed this.

2.32 L393: I suggest to write "… **presumably** contributing to larger diabatic heating and higher temperatures"

**Response:** We have changed this.

2.33 L500: To better connect to the first part of the sentence, you could insert something like "while the contributions of adiabatic and diabatic processes are generally smaller and vary more in space and also between warm and cold events".

**Response:** We have changed this.

2.34 L513/514: Here, scientific debate is still going on. There are also studies saying that heat waves in the mid-latitudes are driven by advection (e.g., Harpaz et al. 2014, Sousa et al. 2019, partly Röthlisberger and Papritz 2023).

**Response:** We have added a more nuanced discussion here.

2.35 L537: I do not understand what is meant by the sentence "is approximated through the average temperatures of the trajectories initiated on the corresponding day at their initiation time".

**Response:** We have added the following explanation: "This is an approximation, as the trajectories are initialized from different heights above the surface, assuming (and sampling) a well-mixed near-surface layer."

**3  Technical correction- Reviewer 1**

**3.1  Title: temperature without plural?**

**Response:** We have changed this

**3.2  L120: 5th and 95th percentiles of the DTDT change distribution**

**Response:** We have changed this

**3.3  L125, 136: no plural, DTDT change event?**

**Response:** We have changed this

**3.4  L190: DTDT variation**

**Response:** We have changed this

**3.5  L193: DTDT variation events**

**Response:** We have changed this

**3.6  L194: DTDT variations**

**Response:** We have changed this

**3.7  Figures: I think it would be helpful to keep the direction of panel labeling consistent across all figures.**

**Response:** We have kept all the panel labels in the vertical direction.

**3.8  L202/284/353/…: Should we invest in the mechanisms?**

**Response:** We have changed this

**3.9  L220: On the days of the ...**

**Response:** We have changed this

**3.10  L241: smaller in magnitude?**

**Response:** We have changed this

**3.11  L339: compared **to**

**Response:** We have changed this

**3.12  L445: result from**

**Response:** We have changed this

**Second round of review of "Physical Processes Leading to Extreme day-to-day Temperatures Changes, Part I: Present-day Climate" by Kalpana Hamal and Stephan Pfahl submitted to Weather and Climate Dynamics**

We would like to thank the reviewers again for their helpful comments. Our responses are printed in blue, whereas the reviewer's questions are in black.

**Comments of Reviewer 1 and Responses**

**A. General comments**

This is the second time I am reviewing the manuscript, and I believe it has substantially improved during the first round of review. In particular, I appreciate that the authors are now more precise in their use of terminology (e.g., "advection," warming/cooling events), which significantly enhances the clarity of the text.

However, in response to some of my previous comments (regarding lines 159-161, twice line 214, Caption Figure 4 in the original manuscript), the authors stated that they had incorporated the suggested changes into the revised manuscript. Unfortunately, these changes do not appear in the current version. I assume this was an oversight, and I would like to encourage the authors to implement these revisions in the next version. I still have a few minor suggestions for improving the text, but overall, I feel the manuscript is close to being ready for publication.

1. L159-161: "Since the magnitude of σ changes can be expressed as a function of …, Figures 1 and 2 show these related quantities." Again, I think this sentence is not properly formulated. The Figures 1 and 2 do not show the other quantities because σ can be expressed as a function of them. It is rather that you decided to show them as they are part of the computation of σ.

   **Response:** The sentence has been revised in the updated manuscript as: "According to equation 5, the magnitude of DTDT changes can be expressed as a function of the standard deviation $\sigma_T$, and lag-1 autocorrelation $r_{1,T}$ of daily mean temperature, which is shown in Figures 1c-f and S1e-l also show these related quantities for DJF and JJA."

2. L214: I was wondering whether the word "distinct" is appropriate.

   **Response**: The sentence has been revised in the updated manuscript as: "Over the 3d leading up to this preceding day, these cold air masses (mean temperature of -21.5°C at -3d) experience a gradual temperature increase (of 5.7°C), with significant adiabatic warming (8.3°C in the mean) due to a strong 100hPa mean descent (Figures 5e-f)".

3. L214: What is meant by "limited" diabatic cooling?

   **Response:** The sentence has been revised in the updated manuscript as: "Some diabatic cooling, likely due to longwave radiation, is indicated by a reduction in θ (by -2.6°C in the mean), constraining the temperature increase (Figure 5g)".

4. Caption Figure 4 in the original manuscript, (Caption Figure 4: You write "selected grid point" but what is shown is a "grid box")

   **Response:** We have replaced "grid point" by "grid box".

**B. Minor comments**

1. L61: The sentence "In contrast, tropical regions typically exhibit weaker temperature advection" suggests a comparison, but earlier in the text, you have not explicitly mentioned that other regions exhibit stronger temperature advection. Consider rephrasing to make the comparison clearer and more logically connected.

   **Response:** Thank you for the helpful suggestion. We have revised the sentence to make the comparison more explicit and logically connected.

   In contrast, tropical regions generally experience much weaker temperature advection compared to the extratropics, and extreme temperature events there are more strongly influenced by local processes such as precipitation, radiation, cloud cover, and surface fluxes (Gough, 2008; Matuszko et al., 2004; Sun and Mahrt,

1995; Dirmeyer et al., 2022). Nevertheless, an accelerated warming of extreme temperatures across tropical land has been observed recently (Byrne, 2021).

2.  L61: I feel that the use of "However" at this point may not be appropriate.

    **Response:** The word "However" is no longer in line 61.

3.  L112: "The approximation in equation (4) is based on ..." instead of "… is associated with"?

    **Response:** We have changed this to "is based on".

4.  L124: I suggest removing the word "previous" from the phrase "previous studies on extreme temperatures," as it may imply that your study also focuses on extreme temperatures, which it does not.

    **Response:** We have removed "previous" from the sentence.

5.  L125: It is not clear to me why the near-surface layer must be assumed to be well-mixed. Could you clarify this point?

    **Response:** Thank you for the comment. The near-surface layer is assumed to be well-mixed to justify using multiple trajectory initialization heights to represent the same air mass. Under typical daytime conditions, surface heating generates turbulence that mixes temperature and moisture within this layer, but this turbulence is not explicitly resolved by the ERA5 wind fields that are used for the trajectory calculations.

6.  L127-129: Think about just omitting the fact that you actually computed 10 day trajectories, although in the end you only needed 3 day trajectories.

    **Response:** Yes, we ultimately used only 3-day trajectories for the primary trajectory decomposition analysis. However, in the density plots, we included the color shading from the 5-day trajectories and the contours from the 3-day and 1-day trajectories to provide a comparative reference for the spatial distribution of air parcel origins. Therefore, we believe it is relevant to mention the computation of 10-day trajectories as well.

7.  L136: Where does the "these" refer to?

    **Response:** In the previous sentence, we mentioned the different locations selected for the study. The term "these" refers to those locations where the Lagrangian temperature decomposition will be performed.

8.  L137: I would try to be consistent with the heading of this subsection, so I suggest instead of "Lagrangian temperature variation decomposition" "Lagrangian temperature variability decomposition".

    **Response:** We have implemented the suggested change.

9.  L186: "… while in the tropics, σ DTDT is lower associated with lower σ T , despite lower r1,T ." I have difficulty understanding this sentence. Think about rephrasing.

    **Response:** Thank you for the suggestion. We have rephrased the sentence as follows: "At the same time, in the tropics, $\sigma_{DTDT}$ is smaller because the standard deviation of daily temperature $\sigma_T$ is low, even though $r_{1,T}$ is also lower".

10. L541-544: "… but advection plays a smaller role, in particular for temperature extremes and heat waves in larger parts of the mid-latitudes". I appreciate that you tried to add a more nuanced discussion here. However, I still feel that it is not correct what is stated here, since the literature is not clear about whether advection really plays a smaller role for warm extremes than for cold extremes. Maybe just apply a more cautios formulation, e.g. "… where advection is sometimes thought to play a smaller role, in particular for temperature extremes and heat waves in larger parts of the mid-latitudes"?

    **Response:** We have made the formulation more cautious: "Comparing these processes associated with extreme DTDT changes with the mechanisms leading to usual temperature extremes (heat and cold waves) indicates similarities in the winter season, when temperature extremes are also strongly affected by advection

in many mid-latitude regions (Bieli et al., 2015; Nygård et al., 2023; Röthlisberger and Papritz, 2023b; Kautz et al., 2022), but larger differences in summer, when extreme DTDT events are still primarily driven by advection, whereas advection is, according to several studies, thought to play a smaller role, in particular for temperature extremes and heat waves in larger parts of the mid-latitudes (Zschenderlein et al., 2019; White et al., 2023, Röthlisberger and Papritz, 2023a)".

11. L576: Where does the "this" refer to?

**Response:** In the revised manuscript, we have replaced the sentence as "This equation contains an approximation, as the trajectories are initialized only once a day (while $\delta_T$ refers to daily average temperatures) and from different heights above the surface, assuming (and sampling) a well-mixed near-surface layer."

**C. Technical corrections**

1. L147: Cross the "was"?

**Response:** We have removed the term was.

**Comments from Reviewer 2 and Responses**

**1. Major comments – Reviewer 2**

1.1 **Use of ERA5 and HadGHCND:** I think the comparison between the DTDT variability in ERA5 and HadGHCND is problematic. As the authors show in their Figure 1, there are large differences between the two data sets that are probably not physical. The reason is likely because HadGHCND interpolates station data to construct a gridded data set which likely smooths out the daily variability (rather than lack of station coverage I think). In my opinion, this makes the HadGHCND data set particularly not suited for the study that you are doing here. That being said, it is true that the authors mainly compare the spatial patterns rather than the absolute values of sigma_DTDT between the two data sets. If you really want to compare the absolute values found with ERA5 with measured data you should probably go directly to station data. For these reasons, I would discourage to show the comparison with HadGHCND in the figures of the main text: the authors can include it in supplementary materials if they really want to do this comparison. In this case they should also discuss the differences between the two data sets more. Moreover, the rest of the paper does not use HadGHCND.

**Response:** We agree with the reviewer that the HadGHCND dataset may have smoothed out the variability due to spatial interpolation and the limited number of stations. This can be verified by comparing the HadGHCND dataset with the Berkeley Earth Surface Temperature (BEST) dataset (Figure R1a-h), which incorporates additional data sources beyond HadGHCND. This comparison shows an increased variability pattern in the northern hemisphere for both DJF and JJA (Figures R1b and d). Furthermore, this allows for a more robust comparison with the ERA5 data for all the quantities (Figure R2).

[Figure]

**Figure R1. (a-d) Standard deviation of DTDT variations ($\sigma_{DTDT}$, °C), (e-h)) standard deviation of daily mean temperature ($\sigma_T$, °C), and (i-l) lag-1 autocorrelation of daily mean temperature ($r_{1, T}$) in December-February (DJF) and June-August (JJA) derived from the HadGHCND and BEST datasets.**

Following the reviewer's suggestion, we have used only ERA5 data for the main manuscript, while this observational analysis has been moved to the supplementary material (Figure S1).

[Figure]

**Figure R2. (a, b) Standard deviation of DTDT variations ($\sigma_{DTDT}$, °C), (c, d) standard deviation of daily mean temperature ($\sigma_T$, °C), and (e, f) lag-1 autocorrelation of daily mean temperature ($r_{1, T}$) in December-February (DJF, 1st column) and June-August (JJA, 2nd column) derived from the ERA5 datasets.**

**1.2 Statistical suggestions**

1.2.1 I think it would be interesting to show (at least for the grid points studied) the distribution of delta_T and the quantiles that you are selecting. In particular it would be interesting to see whether the distribution is symmetric. You could for example compute, in addition t its standard deviation, its kurtosis and show the corresponding map.

**Response:** Thank you for your suggestions. We have calculated the DTDT distribution for each selected location during DJF and JJA, as shown in Figure R3. In DJF, North America exhibits the highest variability with a broad distribution, while South America shows the lowest variability with a sharper peak. Europe and Australia display moderate variability, with intermediate kurtosis values and slight distribution asymmetry. However, in JJA, South America becomes more variable, while North America, Europe, and Australia maintain relatively stable distributions with lower variability than DJF. Additionally, the distributions become more negatively skewed in JJA. We have added this result to the supplementary material (Figure S2).

[Figure]

**Figure R3. Day-to-day temperature (DTDT) distribution curves over the selected locations: North America (black), Europe (orange), Australia (green), and South America (purple) for (a) December-February (DJF) and (b) June-August (JJA). The small dots on the left and right represent the 5th and 95th percentiles, respectively.**

1.2.2    One question I had while reading the paper is how much the extremes of delta_T relate to extremes of T, in other words: do your warming/cooling events also correspond to warm/cool extremes? I think it would be super interesting to show how the extremes of delta_T are linked to the quantiles of T_t and T_t-1. For example, do extreme warming events happen because we start from a very cold quantile and we end up in the middle of the temperature distribution or do we start from the middle of the distribution and end up in the right tail? The physical processes in these two situations are likely different.

**Response:** Thank you for your insightful suggestion. We have illustrated the relationship between DTDT changes and the specific quantiles (terciles) of $T_t$ and $T_{t-1}$ in Figure R4. Our analysis reveals that extreme warming events originate in the lower to middle-temperature quantiles of $T_{t-1}$ and shift toward the middle to higher quantiles of $T_t$. Conversely, extreme cooling events typically begin in the middle to higher quantiles of $T_{t-1}$ and shift to the middle to lower quantiles of $T_t$. We have incorporated this analysis into the manuscript in Section 3.2.

[Figure]

**Figure R4. Heatmaps of the relationship between DTDT change and the deciles of temperature on the previous day (T$_{t-1}$) and the event day (T$_t$) for (a) December-February (DJF) and (b) June-August (JJA) for**

**North America. The x-axis and y-axis represent deciles of $T_t$ and $T_{t-1}$, while the color shading indicates DTDT changes, with red and blue colors indicating warming and cooling, respectively. The black circles represent extreme DTDT changes.**

1.3  **Comparison with climatology:** I find it really interesting that in Figure 4-5 and others the warming and cooling events seem the reverse of one another. As advection seems to be the largest contributor, it seems to me that extremes of sigma_DTDT happens as if this mechanism was switched on or off: warming events happen because the northward advection was switched off and vice versa for cold events. This leads me to my question which is not unrelated to my previous comment 2.a., how are the dynamical situations that you identify unusual with respect to climatology? Is it the starting point that is dynamically unusual or the end point? To be more clear, it seems to me that you should probably do your composite maps also in anomalies.

Response: Thank you for your insightful suggestion. We have analyzed the atmospheric circulation anomalies for the two days involved in an extreme DTDT change event with respect to the seasonal climatology, revealing significant deviations from the mean. Specifically, warming events are associated with southerly wind anomalies and higher geopotential heights (Figure R5), while cooling events are linked to northerly wind anomalies and lower geopotential heights. We added this to the supplementary (E.g., Figures S3 and 6).

[Figure]

**Figure R5. Composite of near-surface temperature anomalies (T2M, °C, color shading), wind anomalies at 850 hPa (UV, m/s, vectors), and geopotential height anomalies at 500 hPa (GP, gpm, magenta contours, dotted and bold magenta contours show negative and positive values, respectively) with respect to seasonal mean on the (a, c) previous day (t-1) and (b, d) event day (t) of the warming (a,b) and cooling events (c,d) during December-February (DJF) at a selected grid point in North America (red grid). Note that wind vector anomalies ≥2m/s are plotted.**

1.4  **Extremes of DTDT and fronts:** the fact that advection is the main factor of extremes in DTDT in the extra-tropics is not super surprising and that is what I would expect because of the existence of atmospheric fronts (some may even argue that fronts are by definition extremes of DTDT). I am surprised that the authors do not

mention at all these structures. Can you say a word about how your analysis and results relate to the literature on frontal structures? Moreover, it seems to me that frontal structures are well identified in the climate variability literature as being the mechanism for day-to-day variability (e.g. Ghil and Lucarini (2020)), maybe you could also mention that in the broader context of your work.

**Response:** We agree that atmospheric fronts play a pivotal role in shaping DTDT extremes in the extratropics. Baroclinic instability drives the formation of frontal structures, which are closely linked to the development of cyclones and anticyclones and serve as key drivers of day-to-day temperature variability (Ghil and Lucarini, 2020). In the composite anomalies of warming events, the transition from cold air masses on the preceding day to warm air masses on the event day corresponds to the passage of warm fronts, which are associated with strong spatial temperature gradients (Figure R5a-b). This phenomenon has been extensively studied and confirmed for European DTDT extremes using different frontal structures (cold, warm, and occluded) (Piskala and Huth, 2020). Our primary objective was to identify the dominant processes driving DTDT extremes. Since our study did not include a database of frontal passages, we initially did not reference this aspect in our analysis. However, we have now incorporated a discussion of this topic, including references to the papers mentioned above, in both the introduction and discussion sections.

**2   Minor comments- Reviewer 2**

2.1 Paragraph L49: in this paragraph you are mainly talking about hot and cold extremes, which are rather different from the extremes in DTDT that you are looking at. This may be confusing for the reader, please be more clear about how the extremes per se relate to the extremes of DTDT (see also my major comment 2).

**Response:** We agree that extreme daily temperature events and extreme DTDT change events are different. However, there is limited literature on the variability of DTDT extremes; we introduce daily temperature extremes to provide the context for atmospheric circulation in the introduction. Furthermore, Equation 5 and Figure R4 illustrate how DTDT variability relates to daily temperature variability. Additionally, our analysis of composites of extreme DTDT and daily extreme events reveals similar circulation patterns, which are, however, more pronounced in the case of daily temperature extremes, as shown in response to Reviewer 1 (their major comment 1). Accordingly, we have updated the introduction and discussion.

2.2  For clarity, I think you should detail a bit more the terms in eq 1-5. In particular, equation 4 is not necessary to me and may be confusing. Moreover, you should explain what the approximation means in equation 3 (explain why this is actually a very good approximation and the errors involved are small because of the typical time scale of the seasonal cycle).

**Response:** We have explained equations 1-5 in more detail in section 2.2.

This study defines DTDT change, denoted as $\delta_T$, as the difference in daily mean near-surface air temperature between the previous day ($T_{t-1}$) and the day of the event ($T_t$), as shown in Eq. (1).

$$\delta_T = (T_t - T_{t-1}) \tag{1}$$

The average daily temperature change, $\mu_{DTDT}$ reflects the difference between the temperatures at the start ($T_0$) and end ($T_n$) of the time series (Eq. 2).

$$\mu_{DTDT} = \frac{1}{n} \sum_{t=1}^{n} (T_t - T_{t-1}) = T_n - T_0 \tag{2}$$

To capture typical day-to-day temperature changes, we thus use the standard deviation, $\sigma_{DTDT}$, as shown in Eq. (3).

$$\sigma_{DTDT}^2 = \frac{1}{n} \sum_{t=1}^{n} (T_t - T_{t-1})^2 \tag{3}$$

By inserting the average daily temperature $\mu_T$ and multiplying out the square bracket, we find a relationship between $\sigma_{DTDT}$, the standard deviation of the daily mean temperature ($\sigma_T$) and the covariance between consecutive days (COV ($T_t, T_{t-1}$)):

$$\sigma_{DTDT}^2 = \frac{1}{n} \sum_{t=1}^{n} ((T_t - \mu_T) - (T_{t-1} - \mu_T))^2$$
$$= \frac{1}{n} \sum_{t=1}^{n} ((T_t - \mu_T)^2 + (T_{t-1} - \mu_T)^2 - 2(T_t - \mu_T)(T_{t-1} - \mu_T)$$

$$\approx 2\sigma_T^2 - 2\text{COV}\,(T_t, T_{t-1}) \tag{4}$$

The approximation in equation (4) is associated with the fact that, for large n, both $\frac{1}{n}\sum_{t=1}^{n}(T_{t-1} - \mu_T))^2$ and $\frac{1}{n}\sum_{t=1}^{n}(T_t - \mu_T))^2$ are good estimators of $\sigma_T^2$. Finally, the standard deviation of DTDT can thus be expressed as a function of the usual standard deviation ($\sigma_T$) and the lag-1 autocorrelation $r_{1,T}$ of daily mean temperature, as shown in Eq. (5).

$$\sigma_{DTDT} = \sigma_T\sqrt{2\,(1\text{-}r_{1,T})} \tag{5}$$

2.3   L126: can you detail a bit more why those choices were made, especially the date and time of the initialization of the backward trajectory.

**Response:** We have added a more detailed explanation in section 2.3:

The Lagrangian analysis tool (LAGRANTO), introduced by Sprenger and Wernli (2015), is used to calculate backward trajectories of near-surface air masses on days associated with extreme DTDT changes from 1980 to 2020. The trajectories are initialized at 18 UTC on both the preceding day   (t-1) and on the event day (t) at 10, 30, 50, and 100 hPa above the surface at the corresponding grid cells. Similar to previous studies on extreme temperatures (Zschenderlein et al., 2019), the different initialization heights are used to sample a near-surface layer that is assumed to be well-mixed. The time difference of 24 hours between the two initializations allows for a proper separation of the air masses before and after the temperature change. Although we use LAGRANTO to calculate 10-day backward trajectories, extremes typically develop on a timescale of 2–3 days (Bieli et al., 2015; Röthlisberger and Papritz, 2023). Therefore, we focus on 3-day backward trajectories for our analysis. Various variables of interest, including latitude, longitude, pressure, temperature, and potential temperature, are interpolated along the trajectory paths and saved at 1-hour intervals.

2.4  Equation 6: this is more for my understanding: given that you are looking at air parcels close to the ground, how can the adiabatic contribution be anything else than positive?

**Response:** While the air masses always undergo adiabatic warming (since they arrive near the surface), the magnitude of this warming can be different, with some air masses descending more than others. Accordingly, the contribution to DTDT changes can be both negative or positive, depending on typical differences in the strength of the descent between the two days. Our results indicate that the mean effect of such changes in adiabatic warming is relatively small in many regions, but they contribute substantially to event-to-event variability.

2.5  Figure 1: the colors scale in all panels is unfortunate. You are showing only positive, non-divergent values therefore you should not use a divergent color maps which is misleading for the reader. Also, because you compare between panels a,b and c,d, the values of the color map should have the same range. Finally, you should probably use the Robinson projection.

**Response:** Thank you for your suggestion. We experimented with a non-divergent color bar but found that the differences in the spatial distribution of DTDT were less clear. To maintain clarity and consistency, we have used the same color bar across all figures, with a similar range for direct comparison. Additionally, we have applied the Robinson projection to improve the representation of spatial patterns (e.g., Figure R1).

2.6   L149 and following: You mention the "magnitude of sigma_DTDT changes." I am not sure what this refers to. If I understand correctly, you should rather talk about the "magnitude of sigma_DTDT."

**Response:** Thank you for the suggestion. We have changed this to "magnitude of $\sigma_{DTDT}$".

2.7  L174: what are "the deep tropics"?

**Response:** Here, we mean to indicate the core equatorial region.

2.8  Because you are studying land grid points only and their proportion varies a lot between the latitudes, I am not sure the zonal means in Figure 1 and following are really relevant: the reader can see by themselves that there is a marked latitudinal gradient of the quantities you are displaying. Also, do you have an explanation

for why in the southern hemisphere the variability is much smaller than in the northern hemisphere for the grid points with the same absolute latitude?

**Response:** Thank you for the suggestion. We have removed the zonal mean representation (e.g., Figure R1).

While, in general, differences in the land-ocean distribution and corresponding spatial temperature gradients may lead to different magnitudes of the variability between the hemispheres (via advection), we do not think that the variability in the southern hemisphere is much smaller than in the Northern Hemisphere at the same latitude. The fact that the maps show a latitude range from 60°S to 90°N might make the comparison a bit difficult. However, we have now extended maps to 90°S and 90°N, and a larger magnitude of DTDT is clearly observed in high latitudes, such as Antarctica (Figure R2).

The variability in the Southern Hemisphere is generally lower than in the Northern Hemisphere at the same latitude in DJF. As shown earlier, advection is the key driver of this variability, and the magnitude is thus related to horizontal temperature gradients. In the Northern Hemisphere, larger land masses are associated with larger temperature contrasts between continents and ocean, while in the Southern Hemisphere, in particular at latitudes south of 40°S, such land-sea differences are much smaller due to the small land fraction and the dominance of oceanic air masses. During JJA, when the meridional temperature gradient is larger in the Southern Hemisphere, the magnitude of the variability becomes more comparable between the two hemispheres.

2.9 L205: "DJF warm events": I would strongly discourage you from using this phrasing for the events you are studying because it is really misleading for the reader. You should talk about warming/cooling events.

**Response:** We have changed "warm/cold events" to "warming/cooling events".

2.10 Fig4: a. Please define more precisely near surface temperature: is it T2M? b. I would suggest not to use absolute values for composite maps: first because there is still the seasonal cycle (how do you handle that by the way?) and second because it is difficult to read and the sudden change of temperature from t-1 to t is not very clear. Maybe simply use anomaly maps and/or make a difference between the map at t and at t-1?

**Response:** Yes, near-surface temperature (T2M) is indeed used, as mentioned in the methodology section. In response to the reviewer's suggestion, we have plotted composite maps for the two days involved in an extreme DTDT change event relative to the seasonal climatology, highlighting significant deviations from the mean (Figure R5). We appreciate the reviewer's recommendation, as the sudden temperature change from t-1 to t is now clearly evident.

Additionally, we plotted the difference between the event day and the previous day, along with the absolute values, to illustrate both the changes and their magnitudes (Figure R6). In the revised manuscript, we have presented each day's circulation patterns and their differences, while the climatology anomalies have been included in the supplementary material.

[Figure]

**Figure R6. Composite of near-surface temperature (T2M, °C, color shading), wind at 850 hPa (UV, m/s, vectors), and geopotential height at 500 hPa (GP, gpm, magenta contours) on the (a, c) previous day (t-1), (b, d) event day (t ) and (c, f) difference of event day and previous day of the warming (a-c) and cooling (d-f) events during December-February (DJF) at a selected grid point in North America (red grid). Note that (in a-d) wind vectors ≥5m/s and (in e-f) wind anomalies ≥1m/s are plotted. The dotted and bold magenta contour in c and f indicates negative and positive geopotential height differences, respectively.**

2.11 Figure 5 and alike: it would be clearer for the reader if you could indicate explicitly on the figure (not only in the legend) if those are DJF or JJA events.

> **Response:** Thank you for the suggestion. We have now indicated whether DJF or JJA events are shown in the Figures themselves (e.g., Figure R6).

2.12 L431: "To systematically investigate the mechanism driving DTDT extremes over the subtropics in the southern hemisphere during DJF and JJA, we select a specific location in Australia": to me this sentence sounds self-contradictory, how can you systematically investigate if you look at only one grid point?

> **Response:** We have improved this sentence structure: "To investigate the mechanism driving extremes of DTDT events over the subtropics in the southern hemisphere during DJF and JJA, we select a specific location in Australia."

2.13 The conclusions reached are based on the analysis of only some grid points at various longitudes/latitudes. Although I think the conclusions reached can probably be extended to the other grid points in the vicinity, I think the authors should be a bit more cautious in their concluding statements.

**Response:** The results for a few additional grid points (Northern Asia, Southern South America, South Asia, Africa, and Western North America) are presented in supplementary material. Nevertheless, we have revised the wording to make it more cautious regarding potential spatial variability.

2.14 L531: "This dominant effect of advection also explains why the magnitude of DTDT changes is typically larger in the extratropics, where horizontal temperature gradients and wind velocities are larger compared to the tropics": this statement is likely true but deserves more evidence.

**Response:** Here, we are not sure which kind of evidence the reviewer would like to see. Based on our composite analysis of atmospheric circulation and 3d backward trajectories, we show that advection dominates DTDT changes in the extratropics. The fact that horizontal temperature gradients and wind velocities are larger in the extratropics than in the tropics is evident from basic climatological data. The relationship between advection, wind velocity, and temperature gradient is clear from the Eulerian version of the thermodynamic equation, where the advection term is written as the scalar product of the horizontal wind vector and temperature gradient. Finally, the magnitude of the adiabatic and diabatic terms in our Lagrangian budgets is of the same magnitude in the extratropics and tropics and thus cannot compensate for the difference in advection. A note on the last point has been added to the conclusion section.

**Second Round of review of "Physical Processes Leading to Extreme day-to-day Temperatures Changes, Part I: Present-day Climate" by Kalpana Hamal and Stephan Pfahl submitted to Weather and Climate Dynamics**

**Comments of Reviewer 2 and Responses**

I thank the authors for taking into account my and the other reviewer's comments. I think these comments have been satisfactorily answered, and I recommend the paper for publication. I would still add a final complain about Figure 1 where the authors argued against my suggestion to use a non-divergent color maps: I am afraid I must insist on this suggestion because I think it can strongly distorts the understanding of the figure by the readers. I left the final choice to the editor.

**Response:** Thank you for your valuable suggestion. We have updated Figure 1 to use a non-divergent color map as recommended to improve clarity and avoid potential misinterpretation (see Figure 1 below). We appreciate your helpful feedback.

[Figure]

**Figure 1. (a, b) Standard deviation of DTDT variations ($\sigma_{DTDT}$, °C), (c, d) standard deviation of daily mean temperature ($\sigma_T$, °C), and (e, f) lag-1 autocorrelation of daily mean temperature ($r_{1,T}$) in December-February (DJF, 1st column) and June-August (JJA, 2nd column) derived from the ERA5 dataset.**